

# Arabic hate speech detection using deep learning: a state-of-the-art survey of advances, challenges, and future directions (2020–2024)

Mariam Itriq* and Mohd Halim Mohd Noor*

School of Computer Sciences, Universiti Sains Malaysia, Pulau Pinang, Malaysia
* These authors contributed equally to this work.

## ABSTRACT

The proliferation of social media has intensified concerns about the societal and psychological impacts of hate speech, particularly in Arabic-speaking communities, where dialectal diversity, morphological complexity, and sociopolitical factors complicate detection. Despite platform efforts, the automated detection of Arabic hate speech remains challenging owing to limited annotated datasets and linguistic nuances. This survey reviews the advances (2020–2024) in deep learning approaches, including convolutional neural networks (CNNs), recurrent neural networks (RNNs), transformer-based models (*e.g.*, bidirectional encoder representations from transformers (BERT) and AraBERT), and hybrid architectures for Arabic hate speech detection. It further examines the dataset constraints involving dialectal variation, annotation inconsistencies, and scarcity. The analysis identified critical research gaps and proposed future directions: expanding multilingual datasets, enhancing contextual modeling, and developing ethically grounded frameworks. This review consolidates state-of-the-art methodologies to guide effective countermeasures against Arabic online hate speech.

## INTRODUCTION

The global Arabic-speaking population, which is around 420 million (*Albadi, Kurdi & Mishra, 2018*), has led to significant adoption of social networks within Arab communities. This trend has resulted in a substantial increase in Arabic digital content (*Abdelali et al., 2021*), with Arab regions producing approximately 27 million tweets per day (*Nagoudi & Abdul-Mageed, 2018*). While social platforms promote enhanced political discourse and organizational engagement, they simultaneously contribute to the spread of harassment and hate speech. Current moderation strategies, which are mainly based on user reports and manual reviews, are becoming inadequate given the high volume of content. Although research on hate speech detection has expanded for languages such as English (*Kovács, Alonso & Saini, 2021*; *Yuan et al., 2019*; *Zampieri et al., 2020*; *Santosh & Aravind, 2019*; *Waseem & Hovy, 2016*); Italian (*Bosco et al., 2018*); Korean (*Founta et al., 2018*); Turkish

Corresponding author
Mariam Itriq,
mariamitriq@student.usm.my

(*Zampieri et al., 2020*); Hindi (*Santosh & Aravind, 2019*). Arabic remains less studied due to unique linguistic challenges, including morphological complexity, lexical ambiguity, and dialectal diversity (*Alruily, 2021*; *Lulu & Elnagar, 2018*). Despite the application of Natural Language Processing (NLP) methods to Arabic (*Aldjanabi et al., 2021*; *Mohaouchane, Mourhir & Nikolov, 2019*; *Albadi, Kurdi & Mishra, 2019*), these challenges continue to hinder the effectiveness of the detection. This survey reviews advances in Arabic hate speech detection (2020–2024), with a particular emphasis on deep learning approaches. Addresses issues related to dataset constraints, model architectures (including convolutional neural networks (CNN), recurrent neural networks (RNNs), and transformers), and performance limitations. The analysis is aimed at natural language processing (NLP) researchers, computational linguists, and platform developers, providing vital details to improve detection frameworks tailored to Arabic-speaking communities.

## Literature search methodology

This study adopted a comprehensive, robust and diversified methodology to evaluate Arabic hate speech detection using advanced deep learning techniques for the period 2020–2024. The research methodology involved extensive searches across prominent academic databases, including IEEE Xplore, Springer Link, ScienceDirect, ACM Digital Library, arXiv, and Scopus. Google Scholar is used to identify additional publications beyond the primary database coverage. Scopus plays a crucial role because of its extensive disciplinary peer-review coverage, facilitating access to high-quality interdisciplinary work in Arabic NLP and hate speech detection. To optimize search efficiency, the research strategy constructs titles, abstracts, and keywords that target relevant materials using the following Boolean query: (TITLE-ABS-KEY ("Arabic hate speech" OR "Arabic offensive language" OR "Arabic abusive language" OR "Arabic cyberbullying" OR "Arabic social media toxicity") AND TITLE-ABS-KEY ("deep learning" OR "neural network*" OR "BERT" OR "transformers" OR "convolutional neural network*" OR "recurrent neural network*") AND TITLE-ABS-KEY ("detection" OR "classification" OR "identification")).

Initial filtration excludes studies outside the scope of the investigation, including the following:

- Non-Arabic language research.
- Traditional machine learning techniques for Arabic hate speech detection.
- Sentiment analysis, machine translation, speech recognition, image processing, and information retrieval studies.

The inclusion criteria prioritize pre-print and peer-reviewed publications (2020–2024) that meet these standards.

1. Focus on handling dialectal variance and multilingual contexts.
2. Direct application of deep learning approaches to Arabic text analysis.
3. Annotation of specific Arabic dialects or regional linguistic nuances.

Supplemental use of Scopus's advanced search parameters identifies foundational publications through document type, publication year, and domain-specific filters.

The analysis categorizes the publications using a deep learning architecture.

- Advanced neural networks (ANNs).
- RNN variants (*e.g.*, long short-term memory (LSTM), gated recurrent unit (GRU)).
- Transformer-based models, particularly Arabic-optimized bidirectional encoder representations from transformers (BERT) variants (AraBERT, MarBERT).

Each study was evaluated according to the following criteria: Methodology refers to the procedures used for architectural design, preprocessing, and training. Annotation techniques, multilingual data integration, and the quantity, variety, and dialectal coverage of the datasets utilized are examples of dataset characteristics. Key performance indicators: The efficiency of the model was evaluated across several activities and languages using key metrics, including accuracy, precision, recall, and F1-scores. The results showed a number of significant developments and trends in the industry: Traditional deep learning designs are giving way to more sophisticated and efficient transformer-based models, such as CNNs and RNNs. These newer architectures allow for improved contextual understanding and adaptability, essential for tackling complex issues like hate speech detection. Modern state-of-the-art systems for hate speech detection across various Arabic dialects and categories have shown promising outcomes, leveraging Arabic-specific versions of BERT, such as AraBERT and MarBERT. These models have been specifically fine-tuned to recognize the nuances in language and sentiment, enabling more accurate classifications of hate speech in diverse linguistic contexts. Despite these advances, challenges remain in the development of large-scale, high-quality datasets that accurately reflect the multitude of dialects and types of hate speech prevalent in Arab-speaking communities. The creation of these datasets is essential to ensure that machine learning models are trained on comprehensive and representative data, which can vary significantly based on regional and cultural factors. To effectively advance the field, it is imperative to address these gaps in data collection and curation. Taken as a whole, this article highlights how deep learning, and more specifically transformer-based models, have revolutionized the field of Arabic hate speech detection. This research offers valuable insights for academics and practitioners, drawing attention to contemporary challenges that require ongoing exploration and identifying potential avenues for further study that could lead to even more effective interventions in combating hate speech online. Consequently, this study addressed these research questions:

- **Q1:** What are the challenges that NLP systems face when dealing with Arabic?
- **Q2:** What are the deep learning models that have been used in the past 5 years to detect hate speech in Arabic, their performance achieved, and the characteristics of the datasets used?
- **Q3:** What are the opportunities available to enhance the models for the detection of Arabic hate speech?

To answer the questions posed above, the remaining portion of this article is set up as follows: An overview of the nature of the Arabic language and the ways it differs from others, in addition to the challenges that arise from these characteristics in detecting hate speech in Arabic, is reviewed in "Background of Arabic Language". "Arabic Hate Speech Detection Datasets" discusses the availability and challenges of building Arabic datasets for hate speech. In addition, it illustrates the generic pipeline and preprocessing steps for Arabic text, and "Hate Speech Detection Approaches" presents the current approaches to detecting hate speech. "Review for Recent Arabic Studies Using Deep Learning" discusses the most recent studies on the detection of Arabic hate speech using deep learning in terms of the models used, the best performance, limitations, future directions, and datasets utilized. Challenges and future directions are presented in "Challenges and Future Directions", while "Discussion" discusses research questions. Finally, the conclusions and contributions of the survey are presented in "Conclusion".

## BACKGROUND OF ARABIC LANGUAGE

More than 422 million individuals, both native and non-native, speak Arabic, making it the fifth most spoken language in the world. In addition, it constitutes the official language of most Middle Eastern countries. Since Arabic is the original language of the Quran and Hadith, Muslims around the world study it in depth. There are three basic varieties of Arabic: dialectal Arabic, which differs from society to society and area to region; classical Arabic, which is the language used in Islamic writings such as the Quran; and modern standard Arabic, which is used as their official language in schools, news outlets, Arabic language publications, and other institutions. According to *Wray (2019)*, the Arab dialects can be divided into five groups. These groups include the Gulf, North African/Maghrebi, Egyptian, Levantine, and Iraqi dialects. The Arabic language is characterized by its richness and complexity at multiple levels morphological, lexical, and orthographic. These unique characteristics present challenges for many NLP applications, including hate speech detection. This section will review the most important characteristics that pose significant challenges for researchers in the NLP field.

### Challenges of Arabic language processing

There are several aspects of Arabic that are challenging for computers to process. According to *Al Chalabi, Ray & Shaalan (2015)*, Arabic has a very complex grammar with several unique elements that are difficult for computers to understand. This richness caused a number of problems that researchers had to approach differently (*Ezzeldin, 2012*). This section addresses a few of these difficulties that can affect the classification of texts, which are one of the basic elements for the detection of hate speech:

**The derivational nature of the Arabic language:** Arabic is a strong derivational language with 10,000 lexical roots of three to four letters each. Lemmas can alternatively be constructed by appending affixes (prefix, infix, or suffix) to each root. For adding affixes to the root, there are 120 distinct patterns. Prefixes can be articles such as (the, ال) prepositions like (from, من) or conjunctions like (and, و). Suffixes such as those used to

indicate the female version of nouns and adjectives, for example, the letter taa' marbootah (ة) can be added to words and adjectives to indicate a feminine form. For instance, the adjective 'generous' manifests as (كريمة) (karīmah) in feminine form *vs* (كريم) (karīm). Furthermore, prefixes and suffixes can be combined, a word can have zero or more affixes. For example, the two words (made مصنوع) and factory (مَصنَع) are formed from the trilateral root (صَنَع); due to this nature of the Arabic language, the classification of Arabic text requires careful preprocessing steps such as stemming, normalization, and tokenization. The following factors may affect how texts are classified due to Arabic's derivational nature:

1. **Word variations:** Arabic words can take on a variety of shapes and variations that all stem from the same root. Prefixes, suffixes, and internal changes are examples of these variants. As multiple forms of the same root may emerge in various contexts, this diversity of word forms can make text classification difficult (*Abdeen et al., 2019*).

2. **The difficulty of stemming:** Stemming is a common preprocessing method in text classifications that involves reducing words to their root form, which is complex in Arabic. It can be difficult to identify the right root form of a word, and incorrect stemming can result in classification problems (*El-halees, 2007*; *Alaa, 2008*).

3. **Semantic relationships:** Arabic words derivational connections can convey significant semantic data. Identifying semantic links between similar terms helps to improve text classification (*El-halees, 2007*).

4. **Feature extraction:** For classification tasks, features that capture the root, prefixes, suffixes, or other derivational patterns can be a valuable source of information (*Azmi & Almajed, 2015*).

**Diacritics:** "Diacritics" are a kind of orthographic symbols that represent a word's sound. In written Arabic, short vowels are denoted by diacritical markings (*Azmi & Almajed, 2015*). To indicate the sound that a letter makes, diacritic marks are positioned either above or below the letter. There is a lot of ambiguity because the same word may have several meanings because modern Arabic writing often lacks diacritics. For example, depending on the context, the word "wrote" "كتَبَ"or "books" " كُتُب" may be used. On the other hand, diacritics can improve the accuracy of text classification models used for Arabic hate speech detection by providing additional information about the pronunciation and meaning of words, for example "غِلّ" it means latent enmity and hatred, but " غُلّ" it means shackles.

**Free word order:** A statement can be written in any order in Arabic: subject-verb-object (SVO), verb-object-subject (VOS), and verb-subject-object (VSO) and still convey the same meaning. Table 1 illustrates an example of Arabic free word order characteristic:

Each sentence in the example is grammatically correct and conveys the same meaning in English: 'The student reads the book'. Variations in Arabic word order can significantly affect the performance of text classification models. Models trained on specific word sequences may become less effective when processing texts with different arrangements.

**Table 1** llustration of Arabic's free word order flexibility using examples of different valid word orders (VSO, SOV, VOS) conveying the same meaning.

| | | | |
|---|---|---|---|
| VSO | الكتابَ the book | الطالبُ the student | قرأ read |
| SOV | الكتابَ the book | قرأ read | الطالبُ the student |
| VOS | الطالبُ the student | الكتابَ the book | قرأ read |

Additionally, models that are fine-tuned for one type of sequence often exhibit reduced performance when used with other sequence types (*Al-Badarneh et al., 2017*).

## Challenges on social media

**Arabizi:** Compared to the many dialects, the Arabic used on social media might be more varied. Transliterated Arabic terms are frequently used in tweets, often written in Latin characters. Arabizi, Arabish, or Franco-Arab are names for this style of Arabic lettering (*Darwish, 2014*). Because not all Arabic letters have corresponding phonetic letters in English, Arabizi uses Latin letters in addition to numerals. For instance, the numbers "2", "3", "5", and "7" stands for the Arabic letters, "أ","ع","خ" and " ح " respectively (*Darwish, 2014*). For example, the Arabizi sentence "2na 3nde 2mt7an al5mees" means "I have an exam on Thursday". In addition, the same Arabizi word may be written in more than one way to give the same meaning. For example, Welcome (مرحبا) can be written in Arabizi spellings (marhaba, mar7ba); due to these factors, it is necessary to classify words as Arabizi or English in context (*Darwish, 2014*).

**Arabic dialects:** Analyzing and processing Arabic text is extremely difficult due to the diversity and variation of Arabic dialects and forms. For instance, the term "lot" is "a lot كثير" in MSA, whereas in Levantine "katheer كثير", in Gulf "maraah مرة" "wayed وايد", in Iraq "ghiwaya غواية", "bezaf بزاف" in Morocco and "yaser ياسر" in Mauritania. Significant dialectal variations exist between regions, sometimes even within the same country, as exemplified by differences observed throughout the Gulf states and Levantine territories, where in some urban areas the knife is called "skeen سكين", while in the Bedouin and rural areas it is called "Khosa خوصة" (*Alhumoud et al., 2015*). Although certain words in various dialects may have the same spelling and sound, they may have different meanings. For example, the Arabic word "Nasih" denotes "overweight" in Levantine, "smart" in Egyptian, and "advisor" in Gulf. As evidenced, the multiplicity of dialects in the Arabic language is a difficult challenge in developing models for text classification in hate speech.

**Ambiguity in Arabic:** Arabic words can have several meanings depending on the context in which they are used, which is known as ambiguity. For instance, the word 'killing' may or may not denote hostility depending on context, exemplified by the maxim: 'You may kill the flowers, but cannot prevent the arrival of spring' تستطيع قتل الأزهار لكن لا تستطيع منع الربيع (*Abdelfatah, Terejanu & Alhelbawy, 2017*).

Arabic's ambiguity can make it difficult to correctly recognize and classify hate speech, which can provide problems for hate speech detection and classification. Furthermore, ambiguity can affect how features are chosen and extracted for text classification. In order

to successfully handle the nuances and context of Arabic text with varied degrees of ambiguity, specialist techniques and models must be developed.

## ARABIC HATE SPEECH DETECTION DATASETS

Recently, researchers have established numerous datasets of Arabic hate speech, which feature extensive collections of labeled tweets and resources that include various dialects and categories. However, these datasets often lack diversity in dialects, categories, and balance among classes. In addition, the limited availability of comprehensive and balanced datasets impedes the development of effective machine learning models that can accurately identify hate speech within Arabic-speaking communities. This section analyzes Arabic hate speech detection datasets, focusing on availability limitations and ongoing research challenges. Furthermore, it details mandatory Arabic text preprocessing techniques and fundamental requirements for detecting and eliminating offensive content.

### Availability and challenges

One of the significant obstacles to addressing hate speech in Arabic is the lack of a comprehensive dataset or *corpus* created for this purpose. This lack of datasets has limited many proposed methods for detecting hate speech in Arabic as compared to English. Among the difficulties and restrictions associated with the Arabic datasets currently available for hate speech identification are:

- **Small dataset size:** The size of many of the Arabic datasets used for hate speech detection may limit the performance of the NLP models trained in these datasets.

- **Lack of diversity:** This is a major limitation in Arabic hate speech datasets, as existing collections frequently fail to encompass the full forms of hate speech that exist in Arabic-speaking nations.

- **Lack of standardization:** It is challenging to create models that can reliably identify hate speech in various dialects of Arabic due to the large regional variations in dialect usage.

- **Limited annotation:** The presence of insufficient annotation in certain Arabic hate speech datasets is a challenge to effectively train NLP models to reliably identify instances of hate speech. The Arabic hate speech dataset has binary labels that distinguish between hate speech and non-hate speech. However, it may not adequately represent the many subtleties inherent in hate speech.

- **Limited studies:** Research on Arabic hate speech detection is lacking, and more work is required in this area. New machine learning methods should be investigated for identifying hate speech in Arabic social media posts, along with the creation of larger and more varied datasets.

- **Standardization in Arabic dialects:** It is challenging to create models that can effectively detect hate speech in many dialects due to the lack of uniformity in Arabic dialects.

Despite these challenges, several initiatives have recently been made to create datasets of Arabic hate speech. For example, in *Mulki et al. (2019)*, 5,812 tweets classified as normal, hateful, or abusive make up a Twitter dataset named Levantine hate speech and abusive

(L-HSAB). The authors of *Mubarak, Khan & Osmadi (2022)* created a novel dataset with two classes for abusive Arabic. The dataset contains 10,000 tweets that have been classified into two distinct classes: offensive (including hate speech and vulgar language) and clean. The annotation adhered to the requirements outlined in OffensEval2019. The researchers conducted an analysis of tweets to identify the dialects, concepts, and genders that exhibit the strongest associations with content that is offensive.

In addition, they provided a comprehensive description of the distinctive features of offensive language in Arabic. The researchers in *Alsafari, Sadaoui & Mouhoub (2020)* built a reliable Arabic textual *corpus* that contains 5,340 tweets with various hate labels. They developed classification models using machine learning and deep learning techniques and assessed their performance within a supervised classification framework. Furthermore, they used datasets with two (clean, offensive, hate), three (clean, offensive, and hate), and six classes (clean, offensive, gender-based, religious-based, nationality-based and ethnicity-based hatred) to train different classifiers. The deep learning model uses mBert features, which produced an F1-score rate of 87.03%, and was the best model for binary classification (abusive/normal). Furthermore, the study by the arHateDetector team (*Khezzar, Moursi & Al Aghbari, 2023*) involved the creation of a sizable dataset of Arabic hate speech from a variety of standard Arabic and dialectal tweets; named arHateDataset consists of 34,000 tweets, 32% of which are hate tweets, while the remaining 68% are normal tweets.

The AraCOVID19-MFH project (*Hadj Ameur & Aliane, 2021*) released a dataset named AraCOVID19-MFH to detect fake news and hate speech related to COVID-19 in Arabic. This dataset contains 10,828 Arabic tweets manually labeled with 10 different labels. A public benchmark dataset called HateXplain (*Mathew et al., 2021*) consists of approximately 20,000 posts that have been annotated from three distinct viewpoints. These points of view include the basic classifications of hate, offensive, and normal. Furthermore, the dataset contains annotations that are based on the target community, which refers to the particular community that has been targeted with hate speech or abusive language in the post. Furthermore, the dataset includes annotations of rationales, which means the specific part of the post upon which the labeling decision is taken. The study reported in *Omar, Mahmoud & Abd-El-Hafeez (2020)* provided a dataset of 20,000 tweets, comments, and posts from several social media sites, including Facebook, Instagram, YouTube, and Twitter, that were manually classified as hate or not. The dataset is used to assess the performance of machine learning and deep learning models. A RNN achieved a maximum accuracy of 98.7%. Recently, *Ahmad et al. (2024)* has undertaken efforts to create a comprehensive Arabic dataset specifically to detect Arabic hate speech. This dataset consists of 403,688 annotated tweets collected from the period between the start of 2014 and the end of 2022. Tweets are classified into four categories: extremely positive, positive, neutral, and negative, based on the presence of hate speech. The dataset specifically targeted the Arabic Jordanian dialect, capturing the unique linguistic and cultural subtleties of this particular region. Furthermore, the recently introduced Saudi Offensive Dialect (SOD) dataset (*Asiri & Saleh, 2024*), consisting of more than 24,000 tweets,

**Table 2 Benchmark Arabic hate speech datasets: sources, reference publications, collection platforms, sizes, and annotation labels.**

| Dataset/link | Ref./Year | Platform | Size | Labels |
|---|---|---|---|---|
| Alakrot A.etal dataset | *Alakrot, Murray & Nikolov (2018a)* | YouTube | 15,050 | Not offensive, Offensive |
| Religious Hate Speech Detection dataset | *Albadi, Kurdi & Mishra (2018)* | Twitter | 5,569 | Hate, Not hate |
| MLMA-hate-speech | *Ousidhoum et al. (2021a)* | Twitter | 3,353 | Disability, Gender, Religion, Sexual orientation |
| L-HSAB-First-Arabic-Levantine-Hate-Speech-Dataset | *Mulki et al. (2019)* | Twitter | 5,846 | Normal, Hate, Abusive |
| OSACT4 Shared Task on Offensive Language Detection (Subtask A and B) | *Hassan et al. (2020)* | Twitter | 10,000 | Task A: OFF, NOT OFF/Task B: HS, NOTH S/ |
| COVID-19-Arabic Tweets-Dataset | *Alshalan et al. (2020)* | Twitter | 975,316 tweet used in *Alshalan et al. (2020)* | Hate (low, average, high), Non hate |
| Dataset Hate speech detection in Arabic Twittersphere | *Alshalan & Al-Khalifa (2020)* | Twitter | 9,316 | Abusive, hateful, normal |
| Multi Platforms Offensive Language Dataset (MPOLD) | *Chowdhury et al. (2020)* | Facebook, Twitter, YouTube | 4,000 | OFF, NOT OFF |
| Fine-Grained H.S Detection on Arabic Twitter | *Ben Nessir et al. (2022), Shapiro, Khalafallah & Torki (2022)* | Twitter | 13,000 | Subtask A: OFF, NOT OFF/Subtask B: HS, NOT HS/Subtask C: Ideology, Religion, disability, race, Social class, and Gender |
| Arabic Hate Speech Dataset 2023 | *Ahmad et al. (2024)* | Twitter | 403,688 | Negative, Neutral, Positive, Very positive |

represents a significant advancement in Arabic NLP. This dataset focuses specifically on the Saudi dialect and is annotated using a hierarchical scheme that ranges from general offensive language to more detailed categories, including general insults, hate speech, sarcasm, and other subtypes of hate speech. This comprehensive annotation strategy improves the utility of the dataset for fine-grained and nuanced analysis of offensive language within Arabic dialects.

Table 2 lists some of the most frequently used data sets in reviewed articles, including the source link of the dataset, reference publication/year, platforms, the size of the data set and the labels used for annotation.

There are several websites and platforms that offer Arabic hate speech datasets for research purposes, such as paperswithcode.com, hatespeechdata.com, GitHub, Kaggle, and Surge AI. Researchers can use these datasets to analyze and understand the patterns and characteristics of hate speech in Arabic, ultimately with the aim of developing effective detection and classification models for it.

## Generic pipeline and preprocessing steps for Arabic text

An essential element to address hate speech on Internet platforms is the creation of a proficient automated detection system. The automatic hate speech detection pipeline has many steps, each of which performs a crucial role in the process of identifying and eliminating offensive content. Data gathering, preprocessing, feature extraction, and

classification are some of these processes. This section first outlines a generic hate speech detection pipeline and details its core stages. Subsequently, it examines preprocessing methodologies specific to Arabic hate speech datasets.

### Generic pipeline for hate speech detection

During the data collection phase, a sizable *corpus* of text data is obtained from several sources, such as social networking sites, forums, and online comments. To eliminate unnecessary or distracting information and standardize the text format, preprocessing comprises cleaning and filtering of acquired data. The preprocessed data are then used to extract useful information using feature extraction approaches such as word frequencies, sentiment analysis, and lexical characteristics. The classification stage, which is the final step, employs machine learning and deep learning methods to train models that can reliably classify the text whether it contains hate speech or not, and it may go further than that through its multi-classification. These stages offer a broad foundation for creating an automatic hate speech detection system, although the actual pipeline may differ based on the application details and the type of data being used. The frequently used stages in the hate speech pipeline are shown in Fig. 1, where:

**Dataset preparation** (*Jahan & Oussalah, 2021*; *Al-Qablan et al., 2023*; *Abro et al., 2020*): The initial stage of the pipeline involves the gathering and preprocessing of the dataset. This procedure frequently involves the collection of data from various social media platforms, followed by cleaning and preprocessing steps, and then the data should be annotated with labels denoting the presence or absence of hate speech in each text. The labeling process can be performed manually or by employing pre-defined rules. These steps eliminate extraneous information and standardize textual content, as discussed in the previous section. Finally, the training and testing portions of the dataset should be separated during data preparation for the next machine learning stage.

**Feature extraction** (*Al-Qablan et al., 2023*; *Abro et al., 2020*; *Awal, Rahman & Rabbi, 2018*): Machine learning algorithms rely heavily on feature extraction to convert raw data into significant numerical representations. Features can be extracted using a variety of methods, such as n-grams, Term Frequency-Inverse Document Frequency (TF-IDF), and bag-of-words (BoW). Using these methods, the algorithm can understand the underlying patterns and relationships within the data. N-grams help capture sequential information by considering adjacent words or characters, while BoW represents the frequency of words in a document, disregarding their order. Based on the frequency of occurrence of a term in a given document and its rarity throughout the dataset, TF-IDF determines the relevance of the term in that document. Using these feature extraction techniques, machine learning algorithms can efficiently process and analyze large amounts of data to make accurate predictions and decisions.

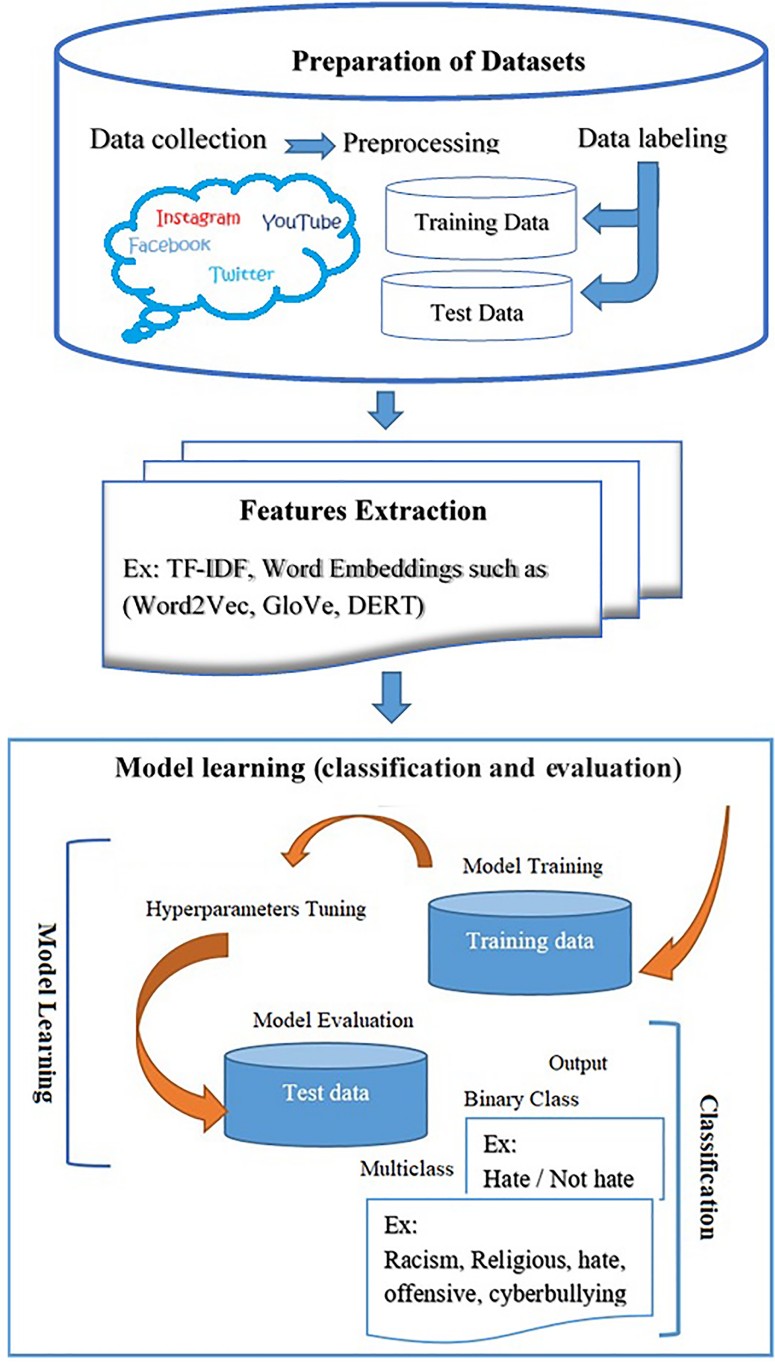

**Figure 1 A comprehensive pipeline for hate speech detection, covering data collection, preprocessing, feature extraction, and classification stages.**

**Model learning (classification and evaluation)** (*Jahan & Oussalah, 2021*; *Muzakir, Adi & Kusumaningrum, 2023*): In this stage, first, an appropriate deep learning or machine learning model is selected, and the chosen model is trained on the training data using the features and labels. The validation set can then be used to improve the model

hyperparameters, such as the learning rate and batch size. Finally, the performance evaluation of the model on the test dataset by employing suitable evaluation measures, such as precision, accuracy, and F1-score, and the recall.

### Preprocessing steps for Arabic hate speech dataset

To obtain clean, standardized, and representative datasets on hate speech, preprocessing procedures are necessary. They permit reliable analysis and modeling for hate speech detection, guarantee consistency in annotation, extract important features, deal with language-specific issues, limit biases, and enhance data quality. The Arabic language differs from other languages in a number of preprocessing steps due to the characteristics mentioned previously. This study addresses preprocessing steps such as tokenization and stemming within a deep learning framework to optimize feature extraction for hate speech detection. The essential steps are reviewed below.

1. **Arabic text normalization:** The goal of normalization is to reduce the number of various letter forms that exist in a single word. For example, the Arabic letters (أ إ آ) are transformed to (ا) another example is transforming (ي) into (ى) and the letter (ة) to (ه). Some models remove diacritics and double the letters that include the symbol (الشدّة).

2. **Cleaning and noise removal** (*Husain & Uzuner, 2022*): Arabic social media posts sometimes include noise, such as hashtags, URLs, emoticons, user mentions, Latin characters, and digits. The quality of the text data can be improved, and more accurate hate speech detection can be facilitated by replacing or removing these noisy portions.

3. **Tokenization** (*Alruily, 2021*; *Awajan, 2007*): The process in which text is split into smaller units, such as words or sub-words, is called tokenization. The absence of distinct word borders in the Arabic alphabet makes tokenization difficult. To tackle this task, Arabic-specific tokenization techniques such as the Farasa tokenizer or the MADAMIRA toolkit are frequently used.

4. **Removing stop words** (*Mubarak, Khan & Osmadi, 2022*; *Khezzar, Moursi & Al Aghbari, 2023*): Eliminating insignificant words can reduce noise levels and increase computing efficiency during subsequent analysis. Some stop words in Arabic are: prepositions (such as: in/في, on/على, to/إلى), pronouns such as: (I/أنا, we/نحن, he/هو, she/هي, they/هم), demonstratives, (this/هذا, these/هؤلاء, there/أولئك), signal words such as: (first/أولا, second/ثانيا), and interrogatives such as:(where/أين, when/متى, whom/لمن).

5. **Lemmatization and stemming** (*Alruily, 2021*): Lemmatization and stemming are two techniques for breaking down words into their root form, Arabic lemmatization and stemming algorithms take into account the complex morphology of Arabic words. This process helps to extract the essential elements of the text data and minimize sparsity.

6. **Part-of-speech tagging** (*Mohamed & Kübler, 2010*; *Alharbi et al., 2019*): This involves labeling each word in Arabic text with its part of speech, such as noun, verb, or adjective, which can help improve the accuracy of NLP models.

**Table 3 Step-by-step examples of key preprocessing phases in Arabic NLP with corresponding input-output transformations and tools used for Arabic text** "هذا الكلام #العنصري و التحريض على الكراهيةِ يجبُ أن يتوقف حالاً."

| Step | Purpose | Example (Input → Output) | Tools/Techniques |
|---|---|---|---|
| Normalization | Standardize text by resolving diacritics, elongations, and dialectal variations. | الكراهيه←الكراهيةِ<br>يجب←يجبُ<br>حالا←حالاً | Python libraries: pyArabic, camel-tools; Rules for MSA/dialect unification. |
| Text cleaning | Remove noise (hashtags, URLs,..) | العنصري←#العنصري | Regex patterns, nltk, custom filters for Arabic social media text. |
| Tokenization | Split text into atomic units (words, subwords). | "هذا الكلام #العنصري و التحريض ..."←<br>"هذا", "الكلام", "العنصري", "والتحريض", | AraBERT tokenizer, Farasa segmenter, or rule-based splitting for clitics. |
| Stopwords removal | Filter out high-frequency, low-meaning words. | "هذا", "الكلام", "العنصري", "والتحريض"<br>←"الكلام", "العنصري", "التحريض", | Custom Arabic stopword lists (MSA + dialects), NLTK Arabic *corpus*. |
| Stemming/Lemmatization | Reduce words to root form. Map words to dictionary base form (context-aware). | العنصر←عنصر<br>يتوقف←وقف<br>يجب←وجب<br>تحريض←حرض<br>حالا←حال<br>كراهيه←كراهي | Khoja's stemmer, ISRI stemmer, or Qalsadi for Arabic. Farasa lemmatizer, CAMeL Tools, or MADAMIRA morphological analyzer. |
| POS tagging | Assign grammatical labels to tokens (noun, verb, *etc.*). | كلام → N (noun)/ عنصر← N (noun)/<br>وجب← V (verb)/ وقف← V (verb)/<br>حرض← V (verb)/ حال← ADV (adverb)/ كراهي → N (noun) | StanfordNLP Arabic model, CAMeL Tools, or UDPipe with Arabic trained data. |

7. **Arabic dialect handling** (*Husain & Uzuner, 2022*): During preprocessing stages, dialect handling methods may be used to account for differences in dialect. These may include standardizing dialect-specific language or using dialect-specific resources.

It is essential to keep in mind that the specific preprocessing procedures may change based on the demands of the hate speech detection task, the characteristics of the data, and the available resources. When addressing the challenges of Arabic text in the context of hate speech detection, preprocessing procedures should be specifically suited to these issues. Table 3 illustrates an example of the main steps in the NLP phase for Arabic text.

Research on Arabic hate speech detection has explored various preprocessing approaches. Table 4 presents a summary of these techniques. Fundamental text cleaning and stopword elimination are prevalent across studies, while normalization methods, such as unifying Alef (ا) to (أ) and Ya (ى) to (ي), are widely adopted to address spelling variations. Furthermore, some researchers implement stemming or lemmatization to reduce word variations, although their impact on classification performance is inconsistent.

## HATE SPEECH DETECTION APPROACHES

Several approaches have been developed to classify hate speech. Some of the several approaches for hate speech detection models are conventional (shallow) and deep learning methods. Deep learning can be based on word embedding or transformer-based (*Chowdhury et al., 2020*; *Ben Nessir et al., 2022*), as shown in Fig. 2.

A brief overview of shallow and deep learning approaches will be given in this section.

**Table 4 Comparative summary of Arabic text preprocessing techniques across selected studies, highlighting common practices and variations.**

| Study | Text Cleaning | Normalization (Alef/Ya) | Tokenization | Stemming/Lemmatization | Stopword Removal | Diacritics Removal |
|---|---|---|---|---|---|---|
| *Ousidhoum et al. (2021a)* | ✓Explicitly mentioned | ✓Explicitly mentioned | Word-based (verified) | ? Unclear | ✓Explicitly mentioned | ✗Not mentioned |
| *Hassan et al. (2020)* | ✓Explicitly mentioned | ✓Explicitly mentioned | Word-based (likely) | ✗Not used | ✓Explicitly mentioned | ✗Not mentioned |
| *Alshalan et al. (2020)* | ✓Explicitly mentioned | ✓Explicitly mentioned | Subword-based (BPE) | ✓Explicitly mentioned (Light Stemmer) | ✓Explicitly mentioned | ✓Explicitly mentioned |
| *Alshalan & Al-Khalifa (2020)* | ✓Explicitly mentioned | ✓Explicitly mentioned | Word-based (likely) | ✗Not used | ✓Explicitly mentioned | ✓Explicitly mentioned |
| *Chowdhury et al. (2020)* | ✓Explicitly mentioned | ✓Explicitly mentioned | Subword-based (WordPiece) | ✗Not used | ✓Explicitly mentioned | ✓Explicitly mentioned |
| *Ben Nessir et al. (2022)* | ✓Explicitly mentioned | ✓Explicitly mentioned | Word-based (verified) | ✓Explicitly mentioned (Khoja Stemmer) | ✓Explicitly mentioned | ✗Not mentioned |
| *Shapiro, Khalafallah & Torki (2022)* | ✓Explicitly mentioned | ✓Explicitly mentioned | Subword-based (Sentence-Piece) | ✗Not used | ✓Explicitly mentioned | ✓Explicitly mentioned |
| *Husain & Uzuner (2022)* | ✓Explicitly mentioned | ✓Explicitly mentioned | Word-based (likely) | ✓Explicitly mentioned (Light Stemmer) | ✓Explicitly mentioned | ✓Explicitly mentioned |
| *Abu Farha & Magdy (2020)* | ✓Explicitly mentioned | ✓Explicitly mentioned | Word-based (likely) | ✗Not used | ✓Explicitly mentioned | ✓Explicitly mentioned |

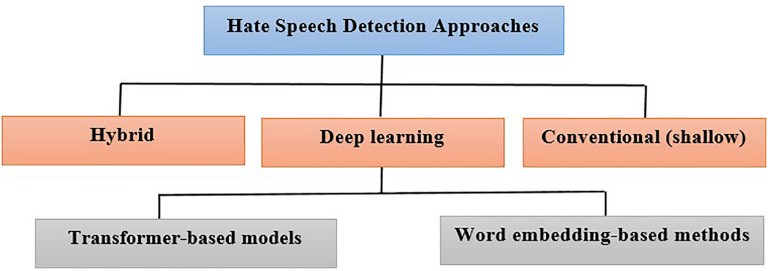

**Figure 2 Taxonomy of hate speech detection approaches (conventional, deep learning, and hybrid) methods.**       

**Shallow methods:** Shallow methods are hate speech detectors that encode text using standard word representation techniques before classifying it using shallow classifiers. TF-IDF (*Aizawa, 2003*) and n-grams (*Davidson et al., 2017*; *Katona, Buda & Bolonyai, 2021*) are two examples of such feature representations that have shown promising results. In terms of classification models, support vector machines (SVM) are among the most widely used classification models for the detection of hate speech (*Muzakir, Adi & Kusumaningrum, 2023*). Additional widely used classifiers for this task are logistic regression (*Davidson et al., 2017*), random forest (*Davidson et al., 2017*), and gradient-boosting decision tree models (*Katona, Buda & Bolonyai, 2021*), which demonstrate efficacy for this task. Instances of hate speech that contain explicit hate speech keywords or phrases can be recognized using shallow methods, which require less computer power than deep learning methods. However, these approaches have

shortcomings in identifying implicitly expressed hate speech, potentially leading to false positives or overlooking edge cases defined by lexicons. The effectiveness of detection models can be improved by integrating shallow methods with complementary strategies such as deep learning.

**Deep learning methods:** Deep neural network-based hate speech detectors are known as deep learning methods. All feature encoding formats, including more conventional ones like TF-IDF, can be used as input data for these neural networks. Using deep learning models, the three most commonly used deep neural network architectures for the detection of hate speech are CNN (*Abu Farha & Magdy, 2020*), long-short-term memory (LSTM) (*Del Vigna et al., 2017*), and bi-directional LSTM (Bi-LSTM) (*Al-Ibrahim, Ali & Najadat, 2023*). Two distinct techniques are employed with deep learning models, which are as follows:

1. **Word embedding-based methods:** Word embedding is a type of NLP technique that creates vectors of real numbers out of words or phrases. This enables the representation of concepts with comparable meaning in a vector space and the identification of semantic connections between words. Numerous word embedding techniques have been developed throughout time, such as word2vec, Glove, and FastText (*Badri, Kboubi & Chaibi, 2022*). Models such as LSTM, Bi-LSTM, and CNN are used in tandem with this method.

2. **Transformer-based models:** By giving researchers powerful tools to capture context, manage distant dependencies, and reach cutting-edge performance, transformer-based methods have transformed NLP jobs. They have established themselves as a pillar in applications involving language and hate speech detection (*Vaswani et al., 2017*; *Awal et al., 2021*). Transformers use self-attentional mechanisms to identify connections among various words in a sequence. This allows the model to focus on the important parts of the input while it is being processed, improving context awareness and the ability to capture long-range dependencies (*Vig, 2019*). Compared to CNNs and RNNs, transformers have better parallelization capabilities. Transformers are now more effective for large-scale NLP problems due to faster training and inference times of this parallelization (*Radford et al., 2019*). In NLP, transformers have been used successfully for transfer learning. Bidirectional encoder representation transformers (BERT) (*Devlin et al., 2019*), GPT ? and RoBERTa (*Tan et al., 2022*) are examples of pre-trained transformer models that have been trained on huge corpora and may be tailored for certain downstream applications, such as hate speech identification.

**Hybrid methods:** This approach involves the integration of many techniques, including machine learning, deep learning, and rule-based methods, with the aim of improving the precision and efficacy of detection models (*Makram et al., 2022*). Table 5 summarizing model types, use cases, Arabic-specific adaptations, and key studies for the detection of Arabic hate speech.

**Table 5 Overview of the main model types for Arabic hate speech detection, examples, use cases, Arabic-specific adaptations, and key studies in Arabic hate speech detection.**

| Model type | Example models | Use case | Arabic-specific adaptations | Key studies |
|---|---|---|---|---|
| **Traditional** | SVM | Binary classification of hate speech using hand-crafted features. | TF-IDF + Arabic lexicons (*e.g.*, hate word lists). | *Alakrot, Murray & Nikolov (2018b)* |
| | Logistic regression | Probabilistic hate speech classification. | Feature engineering for Arabic morphology (*e.g.*, root extraction). | *Ousidhoum et al. (2021b)* |
| | Naive Bayes | Lightweight hate speech detection. | Tokenization tailored for Arabic script and stopword removal. | *Abozinadah, Mbaziira & Jones (2015)* |
| | Random forest | Ensemble-based classification of offensive language. | Dialect-aware feature selection (*e.g.*, Levantine *vs.* Gulf Arabic). | *Aref et al. (2020)* |
| | k-NN | Nearest-neighbor classification for small datasets. | Normalization of Arabic diacritics and elongations. | *Cahyana et al. (2022)* |
| **Deep learning** | CNN | Character/word-level feature extraction for hate speech. | Character-level embeddings to handle Arabic orthography such as (الشدّة). | *Mohaouchane, Mourhir & Nikolov (2019)* |
| | LSTM | Sequential modeling of hate speech context. | Bidirectional LSTM (BiLSTM) for Arabic morphology and word order. | *Al-Ani, Omar & Nafea (2021)* |
| | GRU | Efficient sequential hate speech detection. | Dialect-specific tokenization (*e.g.*, Egyptian Arabic). | *Alshalan & Al-Khalifa (2020)* |
| **Transformers** | AraBERT | Contextual hate speech classification. | Pre-trained on Arabic social media (Twitter) with subword tokenization. | *Khezzar, Moursi & Al Aghbari (2023)* |
| | MARBERT | Dialect-aware hate speech detection. | Pre-trained on 1B Arabic tweets covering multiple dialects. | *Ben Nessir et al. (2022)* |
| | XLM-Roberta | Cross-lingual hate speech detection. | Fine-tuned on Arabic datasets (*e.g.*, OCA, ArSAS). | *Felipe et al. (2022)* |
| **Hybrid** | CNN-LSTM | Combining spatial and temporal features. | Multi-channel input for Arabic dialects and MSA. | *Mohaouchane, Mourhir & Nikolov (2019)* |
| | CNN-GRU | Extract local and sequential features from textual data, GRU captures sequence orders. | | *Al-Hassan & Al-Dossari (2021)* |

# REVIEW FOR RECENT ARABIC STUDIES USING DEEP LEARNING

This section aims to shed light on the current state of research in this field and to identify any gaps or areas that require further investigation. Researchers conducted considerable research on automated hate speech detection in several languages, including English, between 2020 and 2024. However, there is a dearth of studies explicitly focusing on the detection of hate speech in the Arabic language. Figure 3A shows the number of published articles in the field of hate speech detection for Arabic, compared to the number of published articles in the same time period and the same field but dealing with languages other than Arabic, as shown in Fig. 3B. However, there have been notable advances in the use of deep learning models to detect Arabic hate speech. Among them, the use of RNNs, CNNs, GRUs, and BERT is widespread. These models are used both individually and in combination with custom features to improve hate speech detection performance. CNNs have proven useful for text classification applications, such as hate speech detection,

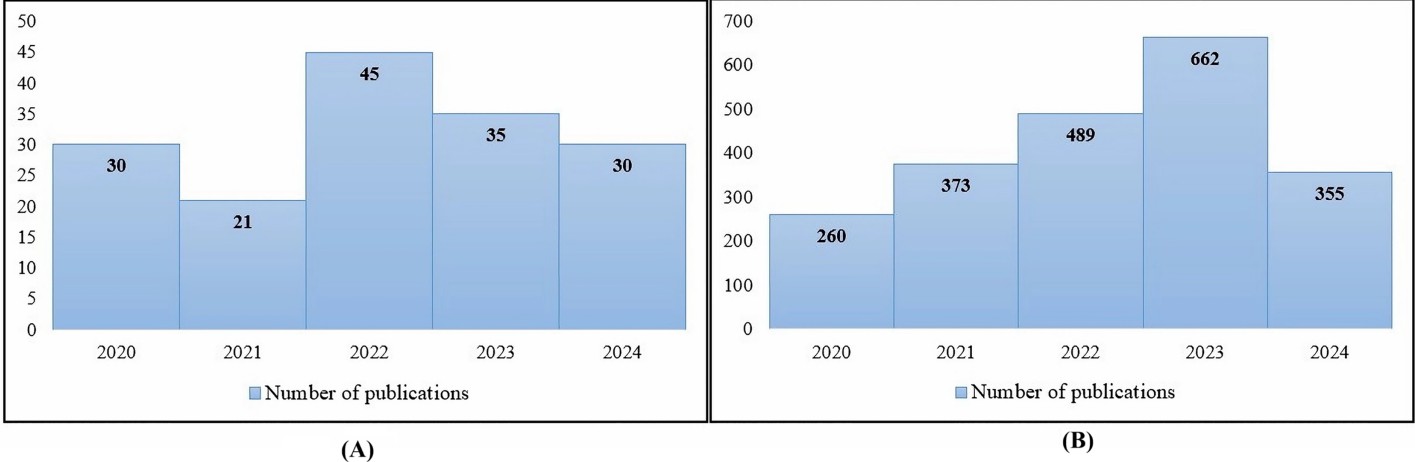

**Figure 3** Annual publications in hate speech detection (2020–2024): (A) Arabic-language focus, depicting emerging research growth; (B) non-Arabic languages, illustrating established research volume (Source: Scopus).

because of their ability to recognize local patterns in input sequences. Their design, which draws inspiration from image processing methods, employs filters to acquire hierarchical representations of textual input, allowing them to analyze the intricate linguistic aspects of Arabic.

GRUs and RNNs, on the other hand, concentrate on identifying temporal connections in text data. Compared to conventional RNNs, GRUs provide a more effective and straightforward model structure that is better suited to managing the vanishing gradient issue. This makes them particularly useful for processing lengthy text sequences, a common occurrence in hate speech. Using transformer models, BERT and its derivatives, such as AraBERT, have completely transformed the domain of NLP. These models rely on self-attention processes to capture dependencies between words in a sequence, regardless of their spatial distance. This ability allows for a more thorough understanding of the subtleties and context of hate speech in Arabic, improving detection accuracy. Furthermore, the development of hybrid models has shown promise in identifying religiously motivated hate speech on Arabic Twitter. These models mix deep learning techniques with manually created characteristics. To increase detection accuracy, these models take advantage of the benefits of both domain-specific knowledge and machine learning approaches.

The following subsections provide a comprehensive overview of progress in this field by analyzing and synthesizing current research conducted on social media platforms for hate speech and abusive Arabic language detection.

## CNN and RNN-based models

CNN is regarded as a network that effectively extracts features from the data. However, RNN works better at modeling jobs with ordered sequences (*Zhang, Robinson & Tepper, 2018*). Some studies test the effectiveness of various deep learning configurations utilizing both RNN alone and RNN in tandem with CNN. Simply, the two systems are more capable

of detecting hate speech patterns. Some of the architectures in the RNN family use different gating techniques; examples include LSTM and the GRU. The researchers in *Mohaouchane, Mourhir & Nikolov (2019)* and (*Abu Farha & Magdy, 2020*) compared various neural network designs for automatically identifying offensive language on Arabic social media. The *Mohaouchane, Mourhir & Nikolov (2019)* models that were put to the test include CNN, LSTM with attention, bidirectional LSTM with attention, and a hybrid of CNN and LSTM. The outcomes show that the CNN model obtains the maximum accuracy and precision, while the CNN-LSTM combination model achieves the highest recall. According to the study, convolutional layers improve model performance, while convolutional layers combined with LSTM layers enhance the recall measure. In their study conclusion, the combined CNN-LSTM model was found to be the most effective for this task. While the most effective model used in *Abu Farha & Magdy (2020)* is a multitask learning architecture built on CNN-BiLSTM that has been trained to predict hate speech, offensive language, and sentiment, the proposed model scored 73.7% on the hate speech test and a 90.4% F1-score in offensive speech. *Faris et al. (2020)* additionally proposed a hybrid CNN-LSTM approach that achieved encouraging results for Arabic hate speech classification on Twitter. The proposed approach achieved the best F1-score of 71.68%. The researchers in *Duwairi, Hayajneh & Quwaider (2021)* conducted training and testing on a substantial Arabic dataset, wherein they observed that the CNN model showed the best results in binary classification with an accuracy rate of 81%. Conversely, the CNN-LSTM and BiLSTM-CNN models yielded the most favorable outcomes in the context of multi-class classification, with an accuracy of 73%. *Al-Hassan & Al-Dossari (2021)* Another study classified Arabic tweets as none, religious, racist, sexist, or general hate. A dataset of 11,000 tweets was annotated. A baseline SVM model was compared to four deep learning models: LSTM, CNN-LSTM, GRU, and CNN-GRU. The findings indicate that the four deep learning models exhibit superior performance compared to the SVM model in the task of identifying abusive tweets. While the SVM has a recall rate of 74% in general, the deep learning models have an average recall rate of 75%. The incorporation of a CNN into the LSTM model results in improved detection performance, achieving a F1-score of 73% and recall of 75%. In a recent publication, *Mazari & Kheddar (2023)* presented a novel dataset specifically designed for the purpose of detecting harmful content in the Algerian dialect. The dataset utilized in this study was obtained from the social media platforms Facebook, YouTube, and Twitter. Numerous experiments have been carried out utilizing several classification models often employed in classical machine learning. Deep learning models, such as CNN, LSTM, GRU, Bi-LSTM, and Bi-GRU, have been extensively evaluated for hate speech detection tasks. The Bi-GRU model has been validated through experimental tests, showing its effectiveness in deep learning classification. It has attained the greatest performance metrics, with an accuracy of 73.6% and an F1-score of 75.8%. In *Al-Ibrahim, Ali & Najadat (2023)*, the authors came up with several deep learning models: the balanced LSTM model, which is similar to the traditional LSTM model but has fewer layers and different parameters in each layer; the improved Bi-LSTM model; the simple GRU model, which is similar to the balanced LSTM model but uses a GRU layer instead of an LSTM layer; the amalgam CNN-LSTM model, which

integrates the characteristics of two deep learning models (LSTM and CNN); and the modified CNN model, which has several layers. The authors of a different study (*Orabe et al., 2020*) used the OSACT4 dataset to look for offensive words. The Word2Vec Arabic embedding let them test a number of deep learning classifiers, such as CNN, CNN with attention, GRU, BiGRU, and BiGRU with attention. The authors incorporated both hostile and non-offensive remarks from an existing Arabic dataset derived from YouTube comments (*Alakrot, Murray & Nikolov, 2018a*). In (*Hassan et al., 2020*), the authors employed a range of traditional machine learning, deep learning, and ensemble classifiers, combining SVM, bagged SVM, and CNN-BiLSTM. The ensemble classifier demonstrated the highest performance among the classifiers, achieving an accuracy of 96.6%, precision of 83.8%, recall of 78.1%, and an overall F1-score of 80.6%. *Haddad et al. (2020)* investigated a Word2Vec Arabic embedding and various deep learning classifiers, including CNN, CNN with attention, GRU, BiGRU, and BiGRU with attention. Additionally, they used an oversampling strategy. The authors incorporated both hostile and non-offensive remarks from an existing Arabic dataset derived from YouTube comments. An F1-score of 85%, accuracy of 91%, precision of 88%, and recall of 83% show that the BiGRU model with attention had the best performance. To improve the system's performance, the authors suggest using LSTM instead of GRU. This would make it easier to find long-range dependencies in the samples. The authors of *Guellil et al. (2021)* proposed three additional distinct machine learning algorithms CNN, LSTM, and Bi-LSTMs for the detection of hate speech against women in the Arabic community on social media sites like YouTube. According to the simulation findings, the CNN model outperforms the LSTM and Bi-LSTM models and obtains the greatest performance, with an F1-score of up to 86% for the imbalanced *corpus*, while the (*Ahmed, Maher & Khudhur, 2023*) researcher employed a combination of CNNs with inverse document frequency GRU and LSTM. In the official deep learning challenge, this ensemble model beat other state-of-the-art models, proving that merging several deep learning architectures may be beneficial. According to the results, when compared to other classifiers, the three-layer inverse document frequency long short-term memory (LSTM) classifier had the highest accuracy rate of 92.75%. Based on CNN, GRU, and BERT, the model proposed by *Alshalan & Al-Khalifa (2020)* was trained on a public dataset of 9,316 Arabic hate tweets using the BERT model and fine-tuned for the task of hate speech detection in the Saudi Twitter-sphere. Experimental results showed that the CNN model performed the best, with an F1-score of 79% and an AUROC of 89%. To utilize the RNN networks, the researcher in *Al Anezi (2022)* proposed models named DRNN-2 and DRNN-1. The algorithm was tested on a unique dataset consisting of 4,203 comments, divided into seven groups: religious, racist, anti-gender equality, violent, insulting, bullying, and neutral. The comments were sourced from different social media platforms. The suggested models achieved an accuracy rate of 84.14% for the seven types of Arabic comments and 99.73% for binary classification. In recent research (*Mousa et al., 2024*), the 1D-CNN, BiLSTM, and radial basis function (RBF) showed effectiveness in detecting Arabic offensive language in Twitter content. These deep learning architectures outperform conventional classifiers like NB and KNN in text classification tasks. BiLSTM networks improve accuracy when used with advanced

embedding methods. Cascading 1D-CNN with RBF yields impressive results, with accuracy, precision, recall, and F1-scores of 88%, 89%, 88%, and 88.2%, respectively. Cascading the sequence (BiLSTM-1D-CNN) with RBF yields the lowest results, with an accuracy of 79%. Table 6 summarizes some recent related works in this context between 2020 and 2024.

## Transformer-based methods

Recent studies have focused on the use of transformers for Arabic hate speech detection. Promising outcomes in a range of NLP tasks, including hate speech detection, have been demonstrated using transformers like BERT and its variants. The ability to understand sentence-level dependencies is the key to the transformer's success because it enables the model to grasp the meaning of a word inside a phrase. This is crucial for the textual hate speech detection task. The model is able to focus on the most important words in a phrase while decoding the tokens that come out of the transformer because of its self-attention mechanism. One notable study that participated in the shared task of hate speech detection at the CERIST NLP Challenge 2022 is *de Paula et al. (2023)*, in which the researchers assessed the performance of six transformer models and their combinations. The CERIST NLP Challenge 2022 organizers claim that AraBERT, AraGPT2, and AraELECTRA were the transformers with the best F1-score performances, which resulted in an F1-score of 60% and an accuracy of 86%. Another study (*Khezzar, Moursi & Al Aghbari, 2023*) introduced a model called arHateDetector, which was trained on a sizable dataset of Arabic hate tweets, arHateDataset, encompassing both standard Arabic tweets and tweets in other dialects of Arabic. A variety of conventional and deep learning machine learning techniques were used to assess the framework's performance. Among these, AraBERT generated the best results with an accuracy of 93% over seven distinct datasets, including the assembled arHateDataset. Furthermore, the study created a website application to assist users in identifying hateful or neutral language in tweets or any other statement. In task 12 of SemEval-2020, many researchers have proposed approaches for detecting both offensive language and hate speech in Arabic. *Tanase, Cercel & Chiru (2020)* describes transformer-based solutions for identifying offensive language on Twitter in five languages, including Arabic. The authors used a number of different neural architectures, such as BERT, mBERT, Roberta, XLM-Roberta, and ALBERT. These were trained using both single-language and multilingual corpora, and they were then fine-tuned using a variety of dataset combinations. Their submission for the Arabic language ranked 28th out of 53 participants in the competition, with an F1-score of 82.19%. While authors in *Socha (2020)* proposed a KS@LTH system that used two models: Arabic BER and XLM-R, achieving 89% F1-scores, in another study on the same shared task, the authors of *Alami et al. (2020)* introduce a technique for utilizing the AraBERT model for identifying offensive Arabic language on Twitter. With an F1-score of 90.17%, the suggested technique performed the best on the OffensEval-2020 Arabic challenge. *Abdul-Mageed, Elmadany & Nagoudi (2021)* presented two reliable deep-bidirectional transformer-based models, dubbed ARBERT and MAR-BERT. These models obtain cutting-edge outcomes on numerous Arabic language understanding tasks after being trained on a variety of Arabic dialects. One of these tasks is

**Table 6 Comparative overview of Arabic hate speech detection studies (2020–2024) using CNN, RNN, and hybrid CNN-RNN models, highlighting preprocessing steps, dataset, model performance, and associated limitations/future research directions.**

| Ref. Year | Preprocessing | Dataset Details | Best model/Performance | Limitations/Future Directions |
|---|---|---|---|---|
| Mohaouchane, Mourhir & Nikolov (2019) | Remove non-Arabic letters, special characters, emoticons, diacritics, punctuation, elongated words. Letter ormalization and tokenization | 15,050 YouTube comments Classes: neutral, offensive, non-offensive | CNN-LSTM Rec = 83.46%, Acc = 87.27% Prec = 83.89%, F1 = 83.65% CNN F1 = 84.5%, Acc = 87.84%, Prec = 86.10% | The study is limited to YouTube comments and may not apply to other platforms. Future research should include Arabic text in Latin alphabet and dialects, and identify other objectionable content on social platforms. |
| Abu Farha & Magdy (2020) | Remove unknown characters, diacritics, punctuation Remove elongation, URLs Normalization | OSACT4: 10,000 tweets Classes: Hate-Speech, Non hate-speech, Offensive, Non-offensive | Hate-speech ( MTL: F1 = 76%) Offensive language (MTL-S-N: F1 = 87.7%) | Risk of mistake transmission with external sentiment data. Need for lexicon augmentation. Multitask learning settings for improved outcomes. Future research will incorporate sentiment data for hate speech and objectionable language. |
| Faris et al. (2020) | Remove non-Arabic characters, symbols, numbers, punctuation, hashtags, web addresses, stop words, diacritics. Normalization. Tokenization | Private/3,696 tweet Classes: Hate, normal | CNN+ LSTM Used AraVec N-grams + SG model with 50 epochs Acc = 66.564% Rec = 79.768% Pre = 68.965% F1 = 71.68% | The model's robustness decreases with smaller datasets, and detecting Arabic hate speech requires larger benchmark datasets and a larger Arabic lexicon, necessitating further research into deep learning methods. |
| Duwairi, Hayajneh & Quwaider (2021) | Removing numbers, English characters, links, hashtags, use mentions, emoji's, and punctuations. Normalizing stripping Arabic diacritics. | ArHS dataset/9,833 tweets Classes: Binary (normal, hate) Ternary (normal, hate, abusive) Multi-class (normal, misogyny, religious, abusive, racism, and discrimination) | Using the ArHS dataset: In the binary CNN F1 = Pre = Acc = 81 In the ternary CNN and BiLSTM-CNN Acc = 74 In the multi-class CNN-LSTM, BiLSTM-CNN Acc = 73 | Previous research on hate speech detection has largely overlooked its complexity, suggesting future studies should expand the ArHS dataset's application to NLP tasks, explore deep learning models, and consider demographic information. |
| Al-Hassan & Al-Dossari (2021) | Remove punctuations, repeated characters, @username, URLs, hashtags.) Normalization. | 11,000 tweets Classes: Religious, Racism, Sexism, General hate speech, Not hate speech | CNN + LTSM Pre = 72% Rec = 75% F1 = 73%. | The study of Arabic tweets faces challenges due to limited data, linguistic diversity, and subjective categorization. Future plans include expanding the dataset, developing real-time hate speech classification methods, exploring alternative text representation techniques, and using advanced hardware. |
| Mazari & Kheddar (2023) | Keeping only Arabic and French characters, deleting numbers, diacritics, elongation, repeated letters more than twice, unknown Unicode, and extra spaces. Substitution of URLs, user mentions, and emoticons with tags and hashtags with separated words. | 14, 150 comments from Facebook, YouTube, and Twitter classes: Hate speech (HS), Cyberbullying (CB), Offensive language (OF). | Bi-GRU Acc = 73.6% F1 = 75.8% | The study of toxic Arabic and hate speech lacks available tools and datasets, and modern NLP methods like BERT, GPT-2, and GPT-3 were not used. Future research should focus on expanding dialectal datasets to include other Arab languages and determining effective word embeddings and methods for dialect toxicity detection. |

(Continued)

| Ref. Year | Preprocessing | Dataset Details | Best model/Performance | Limitations/Future Directions |
|---|---|---|---|---|
| Al-Ibrahim, Ali & Najadat (2023) | Removing Punctuations, links, Numbers, Non-Arabic Characters, Repeated Letters, Arabic Diacritics, spaces and blank lines. Normalization | Private/15,000 tweets Classes: hate/non-hate | Improved Bi-LSTM Acc = 92.20% F1 = 92% | The dataset is accurate but limited to Twitter. Future expansion should include religious, ethnic, political, and hate speech categories, and other platforms such as YouTube. |
| Hassan et al. (2020) | Remove diacritics, words that contain non-Arabic characters, punctuation. Repeated characters replaced with only one. | OSACT4/10,000 tweets Classes: Hate/non hate Offensive/not offensive | Combination of SVMs, CNN-BiLSTM, and M-BERT. Subtask A: F1 = 90.51% Subtask B: F1 = 80.63% | The distribution of hate speech data is uneven, with concerns about vocabulary and word ambiguity. Skewed data makes diagnosing system breakdowns difficult. Future studies aim to identify errors and improve systems by augmenting hate speech data. |
| Haddad et al. (2020) | Removing diacritics, punctuation, non-Arabic characters, emoticons, and stop words, replacing some tokens by their Arabic words. Reducing repeated letters and elongated words. Normalization. | OffensEval 2020/10,000 tweets, and a YouTube comments dataset Classes: offensive, Inoffensive, Hate Speech, Not Hate Speech | Bi-GRU-ATT Offensive (F1 = 85.9% Acc = 91%) Hate speech (F1 = 75% Acc = 95%) | The authors suggest using LSTM models instead of GRU models for improved hate speech detection, as it can better distinguish between hate speech and offensive language, and suggest future research on this approach. |
| Guellil et al. (2021) | Removing diacritics, and Transliteration may be applied to convert Arabizi to Arabic script. Tokenization Normalization. | 5,000 YouTube Comments. Classes: hateful, non-hateful | CNN F1 = 86% | The authors aim to increase the 5,000-comment *corpus* to 10,000 automatically, using automated methods to handle Arabizi, detect Arabic, French, and English comments, and improve categorization performance through transliteration and language identification. They plan to link hate speech identification to sentiment analysis and include a transliteration technique. |
| Al Anezi (2022) | Remove irrelevant data, diacritics, symbols, special characters, and emoji's. | 4,203 comments/not mentioned platforms Classes: Binary (negative/positive) Multi-class (religion, race, gender, violent/offensive, bullying, normal positive, and normal negative) | DRNN-1 Binary Acc = 99.73% DRNN-2 Multi-class Acc = 95.38% | The dataset for hate speech in Arabic could be expanded in size and class count, and machine learning techniques could be improved for higher accuracy. Future plans include creating a prototype system for real-time hate speech monitoring and processing, which can be integrated into social media platforms. |
| Alshalan & Al-Khalifa (2020) | Remove hashtag, stop words and replacing emoji with textual description, punctuation, whitespaces, diacritics, non-Arabic Characters Lemmatization. Normalization. | Private dataset (GHSD)/9316 Tweetsc Classes: hateful, abusive and normal | CNN Pre = 81% F1 = 79% Rec = 78% AUROC = 89% | Hate speech detection is challenging due to tweets, Arabic dialects, and MSA linguistic variations. Future plans include expanding the dataset, adding multi-labels, and testing models on other abusive language datasets, potentially improving classifier performance. |

| Ref. Year | Preprocessing | Dataset Details | Best model/Performance | Limitations/Future Directions |
|---|---|---|---|---|
| Orabe et al. (2020) | Removing repeated consecutive characters, some stopwords, some punctuation marks, and keeping emoji's within the texts | 10,000 tweets classes: offensive (OFF) and not-offensive (NOT OFF) | BI-GRU-ATT Pre = 88% F1 = 85% Rec = 83% Acc = 91% | The dataset has an imbalance between offensive and non-offensive samples, with 81% being non-offensive. To rectify this, the authors plan to use oversampling/undersampling approaches to gather more offensive samples and enhance the dataset with additional abusive and inoffensive tweets. |
| Ahmed, Maher & Khudhur (2023) | Removing special characters, URLs, words in foreign languages, emoji's, and extra spaces. Tokenization. | 32,000 comments/Aljazeera. net Classes: clean, obscene, and rude. | Three-layer LSTM Acc = 92.75%. | High performance but limited to specific domains. Future work focuses on improving deep learning algorithms for Arabic text cyberbullying detection, sentiment analysis, and synthetic methods to protect minors from cybercrime on all social media platforms. |
| Mousa et al. (2024) | Cleaning, normalization, Farasa segmentation, and tokenization. | 13,000 tweets multiclasses: racism, bullying, insult, obscene language, and non-offensive content. | 1D-CNN cascaded with RBF with 88.2% F1-score | This research's limitations include high code complexity from cascaded models, lengthy training periods due to large dataset size, and the need for multiple machine learning model combinations. Future work focuses on improving the cascaded model's performance using models with lower computational complexity. |

social media, where hateful and offensive speech in Arabic is detected. The ARBERT achieved an F1-score of 83% for hate and 90% for offensive, while the MARBERT F1-score was 84.79% for hate and 92.41% for offensive. To detect fake news and hate speech related to COVID-19, the authors of *Hadj Ameur & Aliane (2021)* designed a manually annotated multi-label dataset in Arabic called AraCOVID19-MFH. The dataset has a total of 10,828 Arabic tweets that have been properly annotated with 10 distinct labels, covering several categories such as hate speech, factual information, and others. Several pre-trained transformer models AraBERT, mBERT, DistilBERT-multi, AraBERT Cov19, and mBERT Cov19 were used to evaluate the annotated dataset. The AraBERT Cov19 and mBERT Cov19 models conducted additional pre-training using a dataset consisting of 1.5 million tweets pertaining to COVID-19. For instance, the AraBERT model achieved a weighted F-score of 83.46% without fine-tuning and 98.09% with fine-tuning for the 'Contains hate' task. The result showed that the models pre-trained on COVID-19 data (AraBERT Cov19 and mBERT Cov19) generally performed better than the baseline models. For example, the AraBERT Cov19 model achieved a weighted F-score of 86.49% without fine-tuning and 98.58% with fine-tuning for the 'Contains hate' task, outperforming the baseline AraBERT model. Based on a dataset of comments related to COVID-19, proposed a method using a pre-trained BERT model and deep learning. By combining BERT outputs with GRU and LSTM layers, the (*Chiker, 2023*) authors formed the basis of the proposed approach.

Additionally, the authors propose two approaches to address the class imbalance in the dataset: one utilizes focus loss for training, while the other focuses on data augmentation by oversampling the minority class using the translation and back translation technique. Focus loss training yielded results of 98.03% accuracy and 98.02% F1-score, while data augmentation achieved 99.14% accuracy and F1-score. The researchers in *Boulouard et al. (2022)* conducted a comparative analysis, wherein they examined different approaches such as the translation of comments into English and the utilization of several BERT models. They looked at different methods, such as translating comments into English and using different BERT models. The models employ Arabic YouTube comments as their training dataset. The findings of the study indicate that BERTEN and AraBERT had the best results, with BERTEN demonstrating the highest level of accuracy, precision, recall, and F1-score of 98%. AraBERT was a close runner-up with an accuracy of 96%, a precision of 95%, a recall of 96%, and an F1-score of 95%. *Althobaiti (2022)* proposed a different BERT-based strategy where the authors contrast BERT with traditional machine learning methods and look into the use of sentiment analysis and emoji's as additional features. The experiments show that the BERT-based model outperforms benchmark systems for hate speech detection, offensive language detection, and fine-grained hate speech recognition. The results for the three tasks are: offensive language detection (with an F1-score of 84.3%), hate speech detection (with an F1-score of 81.8%), and fine-grained hate speech detection (with an F1-score of 45.1%). Furthermore, results demonstrate that sentiment analysis marginally improves model performance in detecting offensive language and hate speech, yet exhibits limited impact on classification accuracy for specific hate speech categories. At the fifth workshop on Open-Source Arabic Corpora and Processing Tools (OSACT5), the shard challenge of identifying Arabic abusive language, hate speech, and fine-grained hate speech was presented. Subtask A detected offensive language, which includes explicit or implicit insults or attacks against individuals or groups; Subtask B detected hate speech, which targets individuals or groups based on common characteristics like race, religion, gender, *etc.*; and Subtask C identified fine-grained hate speech. With a new dataset and broader aim objectives, OSACT5, which took place in 2022, could be considered an extension of OSACT4 in 2020. *Mubarak, Al-Khalifa & Al-Thubaity (2022)* provided a detailed overview of OSACT5 shared task in which the teams participating in each task were reviewed, the final results for each of them, and the algorithms used. In their presentation on OSACT5, *Alzu'bi et al. (2022)* proposed aiXplain Inc.'s ensemble-based methodology for identifying offensive speech in the Arabic language. The methodology encompasses many steps, including data preparation, dataset augmentation, and the utilization of an ensemble of classifiers in conjunction with high-level characteristics. The ensemble incorporates many models, namely AraBERTv0.2-Twitter-large, Mazajak pre-trained embedding, character and word-level N-gram TF-IDF embedding, MUSE, and an emoji scoring model. The findings indicate that AraBERT has superior performance, with a precision of 84%, recall of 83%, and F1-score of 84%, and a minor enhancement observed in the overall performance through the utilization of the final ensemble model. The iCompass team (*Ben Nessir et al., 2022*) contributed to OCAST5 with their work on fine-grained hate speech detection on Arabic Twitter. The system has a multi-task model with layers that are tailored to each task

and adjusted with quasi-recurrent neural networks (QRNN). Additionally, a common layer employs cutting-edge contextualized text representation models. The outcomes show that MARBERT improved with QRNN outperformed other models. The iCompass team scored the highest F1 for Subtask B of 83.1% (accuracy = 94.1%, precision = 86.9%, and recall = 80.1%). Additionally, the iCompass team scored the highest F1 for Subtask C of 52.8% (accuracy = 91.9%, precision = 54.8%, and recall = 53.1%). Another submission to OSACT5 was *Shapiro, Khalafallah & Torki (2022)*, in which the authors tested a variety of training methods, pre-processing techniques, and models. They investigated contrastive learning and ensemble approaches, achieving competitive results using MarBERTv2 as the top encoder. The proposed solution achieved 84.1%, 81.7%, and 47.6% macro F1-average in sub-tasks A, B, and C, respectively. By using a combination method of MARBERTv2 and BiGRU models, the researcher of *Bensoltane & Zaki (2024)* achieved a macro F1-score of 61.68% for detecting hate speech in Arabic text. The proposed model outperforms baseline and related work models by combining the BERT model with more intricate neural network layers for encoding context and semantic information relevant to hate speech detection in Arabic. *Almaliki et al. (2023)* introduced the Arabic BERT-Mini Model (ABMM) for detecting hate speech in Arabic tweets on social media platforms. The model utilizes the BERT model to classify tweets into three categories: normal, abusive, and hate speech. The ABMM model achieves high accuracy and outperforms other models in terms of precision, recall, and F1-score with 98.6%.

Additionally, the authors of *Al-Dabet et al. (2023)* and *Masadeh, Davanager & Muaad (2022)* demonstrated that transformer-based models showed promising performance in detecting Arabic hate speech. *Al-Dabet et al. (2023)* used different versions of the CAMeLBERT model and got an F1-score of 83.6% and an accuracy of 87.15%, while *Masadeh, Davanager & Muaad (2022)* used AraBERT and AJGT-BERT to find and label hate speech in religious Arabic, and they did better than traditional machine learning models with an F1-score of 79% for AraBERT and 78% for AJGT-BERT. Recently, in their work (*Alghamdi et al., 2024*), the authors introduce the AraTar *corpus*, an Arabic-language database that has been annotated with various types and targets of hate speech. The authors evaluated several classification models and found that the classical machine learning models SVM and LSTM were inferior to the fine-tuned language models AraBERT in the hate type detection and hate target identification tasks. With better F1-scores of 84.50%, AraBERT models outperformed the competition, proving their efficacy for the detection of hate speech in Arabic text, while the authors of *Zaghouani, Mubarak & Biswas (2024)* selected and annotated 15,965 Arabic tweets from a total of 70,000 tweets, aiming to identify hate speech patterns and establish classification models. The Arabic tweets were annotated: factual accuracy, comedy, rudeness, hate speech, influence on readers, and degree and kind of emotion. The results of the experiment demonstrate that the deep learning technique of the Arabert model yields results with 57% precision and 66% recall for hate speech, 100% precision for non-offensive language and 49% for offensive language. To improve hate speech classification in Arabic tweets, *Eddine & Boualleg (2024)* recommended data augmentation, tweet preprocessing, transfer learning using Arabic BERT models, and ensemble learning. Unlike CNN, LSTM, and the current

deep learning hate speech classification models, they found that the new strategy, which combined ensemble learning and data augmentation, did not require a huge quantity of labeled data to get high performance results. Furthermore, the suggested technique does not require the creation and extraction of handmade features, in contrast to SVM, NB, and bagging. With weighted average F1-score outcomes of 85.48% and 85.10%, respectively, using majority voting and average voting, respectively, the ensemble learning models produce the best results. Using the AraBERT transformer model, *Asiri & Saleh (2024)* research achieved an F1-score of 87% to detect offensive language within the SOD dataset. Performance was further improved to 91% by applying data augmentation techniques designed to address class imbalances in the dataset. These results demonstrate the effectiveness of transformer-based architectures combined with data augmentation in improving offensive language detection in Arabic dialects. The authors of *Mazari, Benterkia & Takdenti (2024)* used deep learning techniques to detect offensive Arabic language, focusing on ensemble frameworks and transformer-based architectures. They used BERT-based variants that were multilingual and dialect-specific, such as QARiB and MARBERTv2, Bi-LSTM networks, and hybrid combinations. Ensemble strategies such as Majority Vote and HighestSum were used for robustness. Data augmentation improved F1-scores, outperforming OffensEval2020 competition winners, with F1-scores as high as 94.56%. Hybrid methods proposed by *Mousa et al. (2024)*, which combine models like BERT with CNN or BiLSTM layers, have greatly improved performance. These cascaded and ensemble architectures have achieved up to 98% in accuracy, precision, recall, and F1-score. Additionally, the inclusion of RBF classifiers within hybrid frameworks has further enhanced results, with the best configurations reaching an accuracy of 98.4%. The authors of *Alabdulrahman et al. (2024)* introduced a dataset of 3,000 Arabic hate tweets, categorizing them into five classes: religion, racism, gender inequality, violence, and offensive content. Traditional methods such as TF-IDF and BoW reached an accuracy of 89%, which was better than the AraBERT model used with SVM and the custom embeddings with RNNs, both of which had an accuracy of 86.53%. Table 7 summarizes some recent related works in this context between 2020 and 2024.

## Comparative analysis and taxonomy of Arabic hate speech detection methods

Researchers are developing various architectures that range from traditional deep learning models to advanced transformer-based systems. Earlier sections have detailed these contributions (see Tables 6 and 7), highlighting a clear distinction between traditional deep learning models, such as CNNs, RNNs, and their hybrids, and transformer-based methods. This section compares and categorizes these approaches to show their strengths, weaknesses, and best uses, as shown in Table 8.

The table presents a comprehensive comparison of the approaches discussed in "Review for Recent Arabic Studies Using Deep Learning", contrasting transformer-based models with conventional CNN/RNN architectures based on several key characteristics. Conventional techniques (such as CNN-LSTM and Bi-GRU) process data sequentially and produce average F1-scores (71.69–92%), but they need a lot of setup and large-labeled

**Table 7 Comparative overview of Arabic hate speech detection studies (2020–2024) using transformers methods models highlighting preprocessing steps, dataset, model performance, and associated limitations/future research directions.**

| Ref. Year | Preprocessing | Dataset details | Best model/Performance | Limitations/Future Directions |
|---|---|---|---|---|
| Tanase, Cercel & Chiru (2020) | Tokenization using BERT specific tokenizer, normalizing hashtag and replacing emoji's with textual description. | Offensive 2020 dataset/ Arabic subset 7000 tweets. Classes: offensive or not offensive | BERT-based model F1 = 82.19% | The authors suggest that the limited availability of training data for non-English languages could improve the performance of multilingual models. They plan to use transfer learning to leverage similar tasks in the same language to enhance offensive language detection models. |
| Socha (2020) | Multiple consecutive user mentions replaced with a single. All tweets truncated or padded to common length. | Dataset in Almaliki et al. (2023)/12,698 tweets. Classes: offensive (OFF)/ not offensive (NOT) | Monolingual: Arabic BERT BASE F1 = 86% | The article does not mention any limitation, challenges or future research directions. Observation: Monolingual models outperform multilingual models for Arabic. A minimal amount of preprocessing was done. |
| Alami et al. (2020) | Substitute emoji's with special token and translate emoji's meanings from English to Arabic, then concatenate emoji's-free tweets with their Arabic meanings. Tokenization. | The Arabic dataset used in OffensEval 2020/10,000 tweet. Classes: offensive (OFF), not offensive (NOT OFF) | AraBERT F1 = 90.17% | The AraBERT model faced a problem with the (MASK0 token not being included in the fine-tuning dataset. The issue was resolved by replacing Twitter emojis with (MASK) tokens. Future work includes using advanced word embeddings. |
| Abdul-Mageed, Elmadany & Nagoudi (2021) | Removing diacritics and replacing URLs, user mentions, and hashtags with generic string tokens (URL, USER, HASHTAG). Tokenization. | For shared task (subtasks A and B) Masadeh, Davanager & Muaad (2022)/10,000 tweet. Classes: social meaning task hate and offensive detection (Hate, not-hate/ offensive, not-offensive) | ARBERT F1 = 83% for hate, F1 = 90% for offensive. MARBERT F1 = 84.79% for hate, F1 = 92.41% for offensive. | The authors aim to improve multilingual language models by self-training and creating models that use less energy, as high inference costs and lack of diversity in non-English pre-training data limit their effectiveness. |
| Hadj Ameur & Aliane (2021) | Removing diacritical marks, links, user's references, elongated and repeated characters. Normalization. Tokenization. | AraCOVID19-MFH dataset/10,828 tweets. Classes: Yes, No, Indeterminate | AraBERTCov19 F1 = 98.58% | The tweet preprocessing was used for model training, not released with the dataset. Authors plan to re-annotate using multiple annotators and expand the annotated dataset with COVID-19 events and discussions. |
| Masadeh, Davanager & Muaad (2022) | Remove punctuation, slang, and stop words. Tokenization. Stemming and Lemmatization. | Dataset used in (Alghamdi et al., 2024)/6,164 tweet+ Arabic Jordanian General Tweets (AJGT) corpus, with 900 tweet. Classes: Hate, Non-hate | BERT-AJGT Acc = 79% | The study focuses on detecting religious hate speech in Arabic, addressing mixed language issues. The future plan is exploring methods for detecting racism, misogyny, and religious prejudice. |
| Boulouard et al. (2022) | Remove emoji's, punctuation, stop words, and extra letters used for emphasis. Some words stemmed and lemmatized. Tokenization. | 11,268 Arabic YouTube comments. Classes: Hateful: 1, Non-hateful: 0 | AraBERT Pre = 95%, F1 = 95%, Rec = 96%, Acc = 96% | BERT models need to handle Arabic dialects, but their versatility limits multilingual performance. Future plans involve using more Levantine and North-African dialect datasets, including "Arabizi". |

(Continued)

| Ref. Year | Preprocessing | Dataset details | Best model/Performance | Limitations/Future Directions |
|---|---|---|---|---|
| *Althobaiti (2022)* | Remove HTML tags, hashtags, mentions, diacritical, punctuation, mathematical signs and symbols, URLs, retweets RT and symbols different from emoji's. Normalization. | The Arabic dataset in *Zaghouani, Mubarak & Biswas (2024)*, consisting of 12,698 Arabic tweets. Classes: Offensive language detection: OFF, NOT OFF. Hate speech detection: HS, NOT HS | BERT-Based Offensive language detection: F1 = 84.3%. Hate speech detection: F1 = 81.8% | The article does not mention any limitation, challenges, or future research directions. Observations: Additional research is needed to properly understand the influence of emojis and their textual explanations, as the dataset used in the study may be too small and uneven. |
| *Alzu'bi et al. (2022)* | Remove URLs, mentions, diacritics, tatweel, punctuation, noisy signals in the tweet. Emoji's translated to Arabic using an English to Arabic model. | OSCAT5 Arabic hate speech task/12,698 tweets. Classes: OFF, NOT OFF | AraBERTv0.2-Twitter-large Pre = 85.2%, F1 = 84.9%, Rec = 84.7%, Acc = 86.4% | Dialect mismatch in pre-trained models makes normalizing tweet dialects, extracting relevant features like POS tags and NER, and recognizing offensive tweets challenging. Future research directions are not explicitly mentioned in the article. |
| *Ben Nessir et al. (2022)* | Remove white spaces, non-Arabic tokens, USER, URL, and emoji's. Normalizing all the hashtags by simply decomposing them. | Dataset in *Zaghouani, Mubarak & Biswas (2024)*/12,698 tweets. Classes: Subtask A: offensive, not offensive. Subtask B: hate, not hate. Subtask C: fine-grained type of hate speech | MARBERT fine-tuned with QRNN Acc = 85.4% for Subtask A, Acc = 94.1% for Subtask B, Acc = 91.9% for Subtask C on the test dataset. | Language complexity in Arabic, cultural, political, and religious dependence, and dialect differences contribute to unbalanced data and class proportions. Future research should explore meta-learning, focus loss, semi-supervised learning, and incorporating disabled and religious minorities. |
| *Shapiro, Khalafallah & Torki (2022)* | Remove repeated characters, emoji's, diacritic and symbols. Normalization. | Dataset in *Zaghouani, Mubarak & Biswas (2024)*/12,698 tweets. Classes: Subtask A: offensive, not offensive. Subtask B: hate, not hate. Subtask C: fine-grained type of hate speech | MarBERT v2 Subtask A F1 = 84.1%, Subtask B F1 = 81.7%, Subtask C F1 = 47.6% | Small or unbalanced dataset overfitting. Larger data sets degrade contrastive loss. Future solutions include using a language-agnostic encoder with contrastive aim and utilizing data from multiple languages for the same function to address data imbalance. |
| *Almaliki et al. (2023)* | Removing @username, URLs, hashtags, punctuation. Tokenization Normalization. | 9,352 tweets. Classes: normal, abusive, hate speech | ABMM (Arabic BERT-Mini Model) Pre = F1 = Rec = Acc = 98.6% | The study suggests incorporating data from Facebook and exploring text representation methods like AraVec to improve neural network model training and enhance the dataset, despite hardware limitations. |
| *de Paula et al. (2023)* | Removing punctuation, special characters, stop words. Converting lower case to upper. Stemming. Tokenization. Lemmatization. | CERIST NLP challenge dataset/10,828 tweets. Classes: Hateful, Not Hateful | AraBERT F1 = 60%, Acc = 86% | Dataset limited to COVID-19 disinformation domain. Small proportion of hate speech in the dataset (11%). The article does not mention any future research directions. |
| *Khezzar, Moursi & Al Aghbari (2023)* | Removing hashtags, stop words, filter out irrelevant symbols. Lemmatization Normalization. | arHateDataset/34,107 tweet. Classes: hate, normal | AraBERT F1 = 93% | Problems with data imbalances and Arabic dialect complexity. The article does not suggest future research directions. |

| Ref. Year | Preprocessing | Dataset details | Best model/Performance | Limitations/Future Directions |
|---|---|---|---|---|
| *Chiker (2023)* | Removing elongations, non-Arabic characters, numbers, symbols, emoticons, punctuation, hashtags, web addresses, empty lines, diacritics. and stop words. Normalization. | Provided by CERIST/10,278 Comments from Twitter and others social media. Classes: Hateful, Not hateful | BERT + GRU and LSTM For focal loss training F1 = 98.02%. For data augmentation F1 = 99.14% | Imbalance between "hateful" and "not hateful" classes. The article does not suggest future research directions. |
| *Alghamdi et al. (2024)* | Removing diacritics, punctuation, repeated characters, symbols, special characters, URLs, English tokens, emoji's. Normalization. | AraTar *corpus*/11,219 tweets. Classes: Task1: RH (Religious Hate), EH (Ethnic Hate), NH (Nationality Hate), GH (Gender Hate), UDH (Undefined hate), CL (Clean) | AraBERTv0.2-twitter (base) F1 = 84.5% | Not all Arabic dialects are incorporated. The future plan is to improve the *corpus* representation for underrepresented hate targets with data augmentation. |
| *Zaghouani, Mubarak & Biswas (2024)* | Removing unwanted characters, English words, and punctuation. | 15,965 tweets. Classes: multi-labels. For hate speech and offensive: Yes, No | AraBERT F1 = 66% for hate speech detection. F1 = 65% for offensive language detection. | Arabic regional backgrounds of annotators may affect labeling accuracy. The article does not suggest future research directions. |
| *Bensoltane & Zaki (2024)* | Removing dates, time, numbers both in English and Arabic, URLs, and Twitter-specific symbols. | OSCAT-5 dataset/12,698 tweets. Classes: offensive, normal, hate (disability, social class, race, gender, religion, ideology). | MARBERT v2+BiGRU F1 = 61.68% | Unbalanced dataset. The future plan is to Combine BERT with different neural network designs and investigate transformer-based models. Find solutions for unbalanced datasets. |
| *Eddine & Boualleg (2024)* | Removing mentions, URLs, RT, hashtags, punctuation, special characters, numerical characters, repeated characters, Arabic stop words, non-Arabic letters, new lines and diacritics. Normalization. | 11,634 tweets +6853 tweets used for data augmentation. Classes: non-hate, general hate speech, sexism, racism, and religious hate speech. | Ensemble learning based on pre-trained models. F1 = 85.48% using majority voting and 85.10% using average voting. Data-augmented model F1 = 85.65% | Confined to specific dataset and time period. The future plan is to improve the contextual embedding model, classify Algerian hate speech, and track trends in hate speech. |
| *Asiri & Saleh (2024)* | replace user mentions with "USER", URLs with "URL", and newline with "NL". | 24,500 tweets, by data augmentation to over 35,000 tweets to address class imbalance. Classes: Offensive/non-offensive. Multi-classes: general insults, hate speech, or sarcasm | 91% F1-score with data augmentation techniques using the AraBERT model | limited regional coverage reduces model generalizability, while models such as AraBERT demand substantial computational resources. Future work should prioritize: (1) developing comprehensive dialect-specific datasets, (2) refining dialect-aware NLP tools, and (3) optimizing models for dialectal variations. |
| *Mazari, Benterkia & Takdenti (2024)* | replace e-mail address and user mentions with <user>, URLs with <url>, and numbers by the ¡number¿, *etc*. Remove Arabic diacritics and elongations, pictographs, symbols, flags, *etc*. | OSACT2020 dataset/10,000 posts Classes: Offensive, Not Offensive | Ensemble learning models based on BERT, 94.56% F1-score. | Class imbalance presents a significant challenge in datasets. Future work will evaluate models for detecting offensive Arabic language forms, explore pretrained BERT variants, and generative AI models to address challenges in detecting such language. |

(Continued)

| Ref. Year | Preprocessing | Dataset details | Best model/Performance | Limitations/Future Directions |
|---|---|---|---|---|
| Mousa et al. (2024) | Cleaning, normalization, Farasa segmentation, and tokenization. | 13,000 tweets Multiclasses: racism, bullying, insult, obscene language, and non-offensive content. | ArabicBERT–BiLSTM-RBF with F score 98.4%. | limitations including computational complexity from cascaded models, extended training times due to large datasets, and reliance on multiple machine learning model combinations. Future work will focus on: (1) adopting faster contextual models to replace BERT architectures, (2) optimizing parameters and feature extraction for efficiency, (3) integrating attention mechanisms for acceleration, and (4) evaluating cross-lingual performance. |

**Table 8 Comparative analysis of recent studies discussed in Tables 6 and 7.**

| Feature | CNN/RNN/Hybrid models | Transformer models |
|---|---|---|
| Architecture | Sequential processing (CNNs: local features RNNs: temporal dependencies) | Self-attention mechanisms (context-aware embeddings) |
| Preprocessing complexity | High (tokenization, normalization, dialect handling) | Moderate (BERT tokenizers handle dialects better) |
| Data efficiency | Requires 10k+ labeled samples | Effective with 5k+ samples (transfer learning) |
| Key strengths | • Interpretable feature extraction<br>• Hardware efficiency<br>• Effective for short texts | • Contextual understanding<br>• Cross-dialect adaptability<br>• State-of-the-art performance |
| Limitations | • Struggles with long-range dependencies<br>• Dialect generalization issues<br>• Platform-specific bias | • Computational intensity<br>• Arabic-specific pretraining needed<br>• Data imbalance sensitivity |
| Top techniques | • Improved Bi-LSTM (92% F1)<br>• CNN-LSTM (83.65% F1)<br>• Bi-GRU (75.8% F1) | • AraBERT (95% F1)<br>• MARBERT (92.41% F1)<br>• AraBERTv0.2 (84.5% F1) |
| Dataset dependencies | YouTube/Twitter-centric (OSACT4, ArHS, OffensEval) | Multi-platform (Twitter, YouTube, Facebook) |
| Fine-tuning | N/A (fixed architectures) | Layer freezing, adapter modules, task-specific pretraining |

datasets (more than 10,000 samples) and have trouble comprehending long-distance connections and various Arabic dialects. Transformer models, such as AraBERT and MARBERT, use self-attention to better understand the context, leading to higher performance (93–99.14% F1) with less data needed (more than 5,000 samples) and greater flexibility with different Arabic dialects. According to the table, transformers are more effective in extracting characteristics and generalizing across platforms, but require more processing power and rely on Arabic-specific pretraining. Important trade-offs are revealed: transformers outperform conventional models in terms of accuracy and dialect resilience, but at a larger resource cost. Conventional models, on the other hand, offer hardware efficiency and interpretability. For practical use in different Arabic language situations, this contradiction shows the need for the best designs that balance computing power and

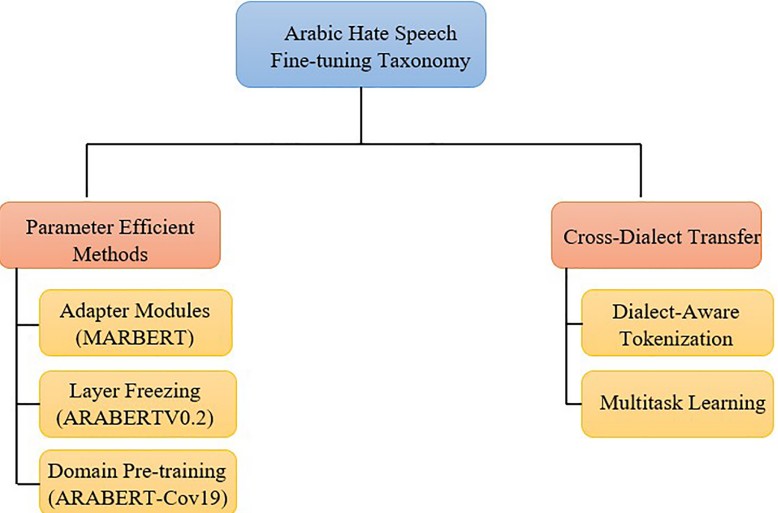

**Figure 4 Taxonomy of fine-tuning techniques derived from Table 8's architectural comparison. Groups methods by their primary optimization goal, with implementations and performance benchmarks from reviewed studies.**

language adaptability. Table 8 further illustrates how transformers utilize structured fine-tuning strategies, such as adapter modules for dialect adaptation and task-specific pretraining for hate speech lexicons, to enhance performance. A visual taxonomy Fig. 4 unpacks these techniques, clarifying their role in achieving cross-dialect robustness.

# CHALLENGES AND FUTURE DIRECTIONS

The challenges in developing powerful NLP models to detect and classify hate speech in all languages, but specifically in Arabic, are among the most crucial issues that need to be addressed and acknowledged while trying to come up with workable solutions or improve what already exists.

## Taxonomy of challenges in Arabic hate speech detection

Arabic hate speech detection presents a complex challenge that spans the linguistic, social, and technical domains. The primary obstacles can be categorized into five different dimensions as shown in Fig. 5:

- **Data availability and quality** (*Khezzar, Moursi & Al Aghbari, 2023*): Labeled datasets for Arabic hate speech are scarce, often focused on specific domains, and exhibit a class imbalance. In addition, they lack diversity in dialects, cultural representation, and consistent annotation guidelines. These shortcomings impact model training, generalization, and evaluation.
- **Linguistic complexity** (*Al-Hassan & Al-Dossari, 2019*; *Darwish, Magdy & Mourad, 2012*): Arabic is morphologically rich, diglossic, and diverse in regions. Hate speech frequently employs dialects, sarcasm, metaphors, and implied meanings, which complicate semantic interpretation for conventional NLP systems.

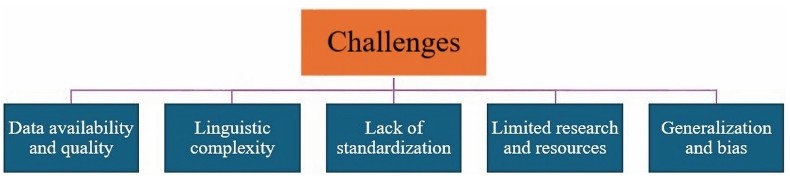

**Figure 5 Taxonomy of challenges in Arabic hate speech detection.**

- **Lack of standardization** (*Darwish, Magdy & Mourad, 2012*): The lack of standardized orthography across dialects complicates preprocessing and annotation. Differences in tokenization, normalization, and transliteration result in inconsistencies that hinder the portability and reproducibility of the model.
- **Limited research and resources** (*Khezzar, Moursi & Al Aghbari, 2023*): Compared to English, Arabic NLP research is limited in terms of available open-source tools, pre-trained models, and shared benchmarks. The use of private datasets in many studies restricts reproducibility and collaboration.
- **Generalization and bias** (*Yin & Zubiaga, 2021*): Models that are trained on narrow, biased, or unbalanced data may fail to perform in real world situations. Automated systems can unintentionally reinforce biases related to gender, religion, region, or politics.

## Future research directions

Addressing the challenges outlined above requires focused strategies and cooperative research efforts. The following recommendations are proposed:

- **Multidialectal and cross-domain datasets:** Construct large-scale, balanced datasets that represent MSA, various dialects, Arabizi, and multilingual content sourced from diverse platforms such as Twitter, Facebook, and YouTube. Ensure the datasets are annotated using robust and standardized guidelines.
- **Real-time detection systems:** Explore low-latency models that are optimized for real-time deployment. Use knowledge distillation to compress large transformer models, such as AraBERT-Mini, to facilitate practical applications.
- **Explainable and ethical AI:** Integrate explainability frameworks to interpret predictions and ensure transparency. Prioritize fairness, data privacy, and ethical design to prevent the reinforcement of societal biases.
- **Bias mitigation and debiasing strategies:** Apply adversarial training and explore methods to address data imbalance, including oversampling, undersampling, and advanced data augmentation. Additionally, utilize fairness-aware objectives to mitigate biases present in models and datasets.
- **Context-aware and culturally informed models:** Develop models that integrate context, such as conversational threads, events, and user history, along with cultural cues. Incorporating societal norms and discourse patterns enhances detection accuracy.

- **Preprocessing techniques:** Working to improve the preprocessing techniques improves the quality of the dataset and thus enhances the performance of hate speech detection models.

## DISCUSSION

This survey presented an overview conducted from the Google Scholar and Scopus digital library databases for most documents related to the detection of Arabic hate speech using deep learning models published between 2020 and 2024. In addition, through this review, many topics related to Arabic language challenges are discussed. Returning to the initial research questions raised, the following paragraphs provide comprehensive and detailed responses to each of them.

According to **Q1: What are the challenges that NLP systems face when dealing with Arabic?**

"Background of Arabic Language" came up with conclusion that the detection and reduction of hate speech in Arabic pose a significant and urgent challenge, necessitating the development and implementation of novel approaches and strategies. The complex characteristics of the Arabic language introduce an additional level of challenge to this task due to its variety of nuances, regional dialects, and cultural allusions that are prone to misinterpretation. Consequently, a comprehensive understanding of the language and contextual nuances is needed in order to efficiently detect and address occurrences of hate speech. Furthermore, the occurrence of hate speech on digital platforms presents an additional challenge in terms of identifying and addressing such content, as it necessitates ongoing surveillance and adaptation to keep up with the ever-changing methods of communication. Another issue is the limited availability of comprehensive Arabic hate speech datasets and the difficulties associated with building new datasets. "Arabic Hate Speech Detection Datasets" presented a set of available Arabic hate speech datasets and reviewed the most important challenges with proposed solutions.

In response to **Q2: What are the deep learning models that have been used in the past 5 years to detect hate speech in Arabic, their performance achieved, and the characteristics of the datasets used?**

Numerous researchers have used deep learning methodologies to detect and classify instances of hate speech in Arabic text. "Review for Recent Arabic Studies Using Deep Learning" provides a comprehensive overview of the current landscape in Arabic hate speech detection, highlighting both advancements and challenges in the field. By examining various models and their effectiveness, the authors set the stage for possible improvements and innovations moving forward. Various models, including CNN, RNN, LSTM, and BiLSTM, have been proposed. Moreover, some articles have indicated that the amalgamation of two or more deep learning models yields superior performance compared to the utilization of a single deep learning model. Both the combined models of CNN+LSTM and CNN+GRU demonstrated superior performance compared to the individual applications of LSTM and CNN. These methodologies have produced promising results in relation to precision and handling complex linguistic structures in the Arabic textual context. Furthermore, current research in the field of deep learning has

placed emphasis on the utilization of transformers, such as BERT and its variants, AraBERT and MARBERT, in the context of Arabic hate speech detection. Furthermore, significant effort has been made to build data sets specifically for the detection of hate speech in the Arabic language, which is crucial to effectively address the issue in Arabic speaking communities. These datasets are created by collecting and annotating Arabic text from various internet platforms; it is noted that interest in the Twitter platform had the largest share. The availability of such datasets enables researchers and developers to train machine learning models that can accurately identify and combat hate speech in Arabic online spaces. The deep learning models with the datasets used are addressed in detail in "Arabic Hate Speech Detection Datasets".

Finally, "Challenges and Future Directions" answered research question **Q3: What are the opportunities available to enhance the models for Arabic hate speech detection?**

The future direction and potential opportunities in the development of Arabic hate speech detection techniques were discussed. The section highlighted the need for further research to refine existing models and explore new approaches to address the evolving nature of hate speech in Arabic. In addition, it emphasized the importance of collaboration between researchers and experts in Arabic language to overcome the issues of building Arabic datasets. In conclusion, the detection and reduction of hate speech in Arabic pose a significant and urgent challenge that requires the development and implementation of new approaches and strategies. The complex nature of the Arabic language, the limited availability of comprehensive datasets, and the difficulties associated with detecting hate speech on social media platforms provide distinct obstacles. However, the utilization of deep learning algorithms shows the potential to tackle these obstacles and drive the detection of hate speech in Arabic to new levels. Researchers have the potential to make significant contributions to combating hate speech in the Arabic language by employing advanced models, utilizing large datasets, and taking ethical factors into account. The application of real-time detection systems, the integration of contextual information, and the use of multilingual techniques can all improve future research and development. Through the use of deep learning techniques and the adoption of a collaborative methodology, it is possible to provide a foundation for online environments that are both safer and more inclusive for groups who speak Arabic.

## CONCLUSION

This survey offers a comprehensive review of Arabic hate speech detection using deep learning models. The analysis highlights the challenges in processing Arabic text, including morphological complexity, dialectal variations, and dataset limitations. Although architectures such as CNNs and RNNs have been applied, transformer-based models such as AraBERT and MARBERT exhibit superior performance in detecting hate speech within Arabic social media content.

This review reveals significant advances in the field, although challenges persist, particularly regarding dataset availability, contextual understanding, and dialectal coverage. Future research should prioritize expanding dataset resources, enhancing contextual awareness in models, and investigating multimodal and cross-lingual

approaches to improve Arabic hate speech detection. Collaboration between NLP researchers and Arabic linguists is essential to address data scarcity and dialectal barriers. In addition, ethical considerations must inform the development of inclusive and impartial models. Addressing these gaps can contribute to safer and more inclusive online environments in Arabic-speaking communities.

### Funding
The authors received no funding for this work.

### Competing Interests
The authors declare that they have no competing interests.

### Author Contributions
- Mariam Itriq conceived and designed the experiments, performed the experiments, analyzed the data, prepared figures and/or tables, authored or reviewed drafts of the article, and approved the final draft.
- Mohd Halim Mohd Noor conceived and designed the experiments, prepared figures and/or tables, authored or reviewed drafts of the article, and approved the final draft.

### Data Availability
This is a literature review and did not utilize raw data.

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
