# Peer review of "Arabic hate speech detection using deep learning: a state-of-the-art survey of advances, challenges, and future directions (2020–2024)"

_PeerJ Computer Science, doi:10.7717/peerj-cs.3133_

## Round 0.1 · original submission · Major Revisions

Reviewer 1 ·

Basic reporting

• Avoid non-academic words such as "good", "well", "notably" and "meaningful" which imply subjective value judgements. Instead, use more precise terms or provide specific metrics where possible.

• Avoid using pronouns such as "we" and "our, " as well as past and future tenses. Academic writing typically uses the present tense in a third-person perspective for formality and clarity.
e.g: - In this study, we discuss preprocessing steps such as tokenization..
- This study discusses preprocessing steps, such as tokenization….

• Avoid using vague verbs like "serve" Instead, choose more specific and descriptive verbs to clarify the function or purpose in context.

• Informal connectors like "Also" should be replaced with more appropriate academic alternatives such as "Furthermore" or "Moreover".

• Refrain from informal expressions such as "is done" or contractions like “can’t”, “it’s”.

• Rewrite the abstract, introduction, and conclusion to ensure they are concise, coherent, and consistent with academic writing standards.

• In survey papers, the phrase “our findings indicate that the field…–in the Conclusion section-” may imply the presentation of new experimental results. To maintain clarity, consider rephrasing to terms such as “our analysis,” “this review reveals,” or “we observe,” .

• The manuscript should also incorporate recent studies—such as those listed below—to ensure broader coverage of current work in the field. Including these references will help provide a more comprehensive and up-to-date survey:
- Alabdulrahman, M., Alotaibi, L., Latif, G., & Alghazo, J. (2024, December). Arabic Offense Text Detection in Social Networks using Collaborative Machine Learning. In 2024 6th International Symposium on Advanced Electrical and Communication Technologies (ISAECT) (pp. 1–7). IEEE.
- Mazari, A. C., Benterkia, A., & Takdenti, Z. (2024). Advancing offensive language detection in Arabic social media: A BERT-based ensemble learning approach. Social Network Analysis and Mining, 14(1), 186.
- Mousa, A., Shahin, I., Nassif, A. B., & Elnagar, A. (2024). Detection of Arabic offensive language in social media using machine learning models. Intelligent Systems with Applications, 22, 200376.
- Asiri, A., & Saleh, M. (2024). SOD: A Corpus for Saudi Offensive Language Detection Classification. Computers, 13(8), 211.

Experimental design

The manuscript aligns well with the journal’s Aims and Scope. It offers a rigorous and well-structured review of the literature, conducted to a high technical and ethical standard.

Validity of the findings

no comment

Reviewer 2 ·

Basic reporting

The manuscript is well-written and professionally presented, with clear language and appropriate structure throughout. The literature review is thorough, and the authors provide sufficient background and context for readers to understand the evolution of fine-tuning techniques for large language models. Figures and tables are helpful, though the captions could be slightly more descriptive to guide the reader without referring back to the main text.

Experimental design

The study design is sound, and the review aligns well with the journal’s scope. The coverage of methods is extensive, and the organization is logical. That being said, a more explicit summary or visual taxonomy of the fine-tuning techniques would help. This will particularly be helpful for the readers to grasp the concept in categorization. Including a brief comparative table of the methods discussed could also enhance clarity and utility.

Validity of the findings

The conclusions appropriately reflect the review’s goals and are grounded in the surveyed literature. The discussion of challenges and future directions is relevant, though a bit more structure in this section, maybe through thematic grouping, would improve readability.

Additional comments

Please refer above.

---

## Round 0.2 · accepted · Accept

Please review your article one more time and follow the next steps for publication.

Reviewer 1 ·

Basic reporting

-

Experimental design

-

Validity of the findings

-